# Hyphal compartmentalization and sporulation in *Streptomyces* require the conserved cell division protein SepX

Matthew J. Bush [1], Kelley A. Gallagher[1,2], Govind Chandra [1], Kim C. Findlay [3] & Susan Schlimpert [1✉]

Filamentous actinobacteria such as *Streptomyces* undergo two distinct modes of cell division, leading to partitioning of growing hyphae into multicellular compartments via cross-walls, and to septation and release of unicellular spores. Specific determinants for cross-wall formation and the importance of hyphal compartmentalization for *Streptomyces* development are largely unknown. Here we show that SepX, an actinobacterial-specific protein, is crucial for both cell division modes in *Streptomyces venezuelae*. Importantly, we find that *sepX*-deficient mutants grow without cross-walls and that this substantially impairs the fitness of colonies and the coordinated progression through the developmental life cycle. Protein interaction studies and live-cell imaging suggest that SepX contributes to the stabilization of the divisome, a mechanism that also requires the dynamin-like protein DynB. Thus, our work identifies an important determinant for cell division in *Streptomyces* that is required for cellular development and sporulation.

[1] Department of Molecular Microbiology, John Innes Centre, Norwich Research Park, Norwich NR4-7HU, UK. [2] Département de Microbiologie, Infectiologie et Immunologie, Université de Montréal, Pavillon Roger-Gaudry, 2900, boulevard Édouard-Montpetit, C.P. 6128, Succursale Centre-ville, Montréal, QC H3C 3J7, Canada. [3] Department of Cell and Developmental Biology, John Innes Centre, Norwich Research Park, Norwich NR4-7HU, UK. ✉email: susan.schlimpert@jic.ac.uk

**M**ost bacteria divide by binary fission which results in the generation of two virtually identical daughter cells that physically separate to colonize new environmental niches. However, in filamentous growing organisms, such as the abundant soil bacteria of the genus *Streptomyces*, two functionally distinct modes of cell division exist, cross-wall formation in vegetative hyphae and sporulation in aerial hyphae (Fig. 1a). Cross-walls are division septa that partition growing hyphae at irregular intervals into multinucleoid compartments, reminiscent of fungal hyphae. In contrast, sporulation-specific cell division results in the formation of ladder-like arrays of division septa that are structurally distinct from cross-walls, and which lead to cell fission and the release of unigenomic spores[1]. While sporulation is an important strategy for survival and propagation[2,3], the physiological significance of cross-wall formation during vegetative growth is much less understood.

At the heart of both division processes is the almost universally conserved cell division protein FtsZ that assembles into a contractile ring-like structure called the Z-ring. The Z-ring acts as a scaffold for the assembly of the cell division machinery (the divisome) and guides the synthesis of septal peptidoglycan[4,5]. *Streptomyces* encode several orthologs of the core divisome components from *E. coli* or *B. subtilis*, such as FtsK, FtsQ, FtsL, DivIC and the cell wall synthetic enzymes FtsI and FtsW[6,7]. Moreover, additional factors have been identified that associate with the divisome and/or contribute to the efficient assembly and stability of Z-rings. These proteins include the actinobacterial-specific protein SsgB, which has been reported to mark future sporulation-septation sites[8], SepF, which is likely to function as a membrane anchor for FtsZ[9–11], SepH, which stimulates FtsZ filament assembly[12], the two dynamin-like proteins DynA and DynB, which stabilize FtsZ-rings during sporulation, and two SepF-like proteins of unknown function[11].

Previous genetic studies have shown that none of the conserved and species-specific cell division proteins are essential for growth and viability in *Streptomyces*[6,7]. While the deletion of core cell division genes, including *ftsQ, ftsL, divIC, ftsW* or *ftsI*, appears to largely affect sporulation septation[6,8,11–13], a *Streptomyces* Δ*ftsZ* mutant is both unable to sporulate and to compartmentalize the vegetative mycelium, resulting in a single-celled, branched mycelial network. Hence, to date *ftsZ* is the only known determinant of cross-wall formation.

Here, we report the identification of SepX, a *Streptomyces* divisome component that is required for compartmentalization of vegetative hyphae and for normal sporulation in aerial hyphae. Besides *ftsZ*, *sepX* is the only other determinant of vegetative septation identified so far and we show that cross-wall formation is crucial for fitness and cellular development in *Streptomyces*. We further demonstrate that SepX also plays an important role in sporulation-specific cell division via direct interaction with the bacterial dynamin-like protein DynB, which was previously shown to contribute to Z-ring stability. We find that while either of the proteins is dispensable for sporulation, deletion of both *sepX* and *dynB* impairs stable Z-ring formation and function, and consequently abolishes sporulation-specific cell division.

## Results

**SepX co-localizes with FtsZ at division septa.** In *Streptomyces venezuelae*, the initiation of sporulation-specific cell division is co-controlled by the WhiA and WhiB transcription factors[14,15]. Using chromatin immunoprecipitation (ChIP)-seq in concert with microarray transcriptional profiling, we previously revealed the WhiAB-mediated activation of several genes encoding core members of the bacterial division machinery, including *ftsW, ftsK* and *ftsZ*. WhiA and WhiB together control the expression of ~240 transcriptional units and we reasoned that other genes encoding key components of the *Streptomyces* division machinery might also be under WhiAB control. Therefore, we constructed various strains expressing fluorescently tagged WhiAB-target proteins, prioritizing those encoded by conserved and uncharacterized genes with a clear dependence upon WhiA and WhiB for their expression.

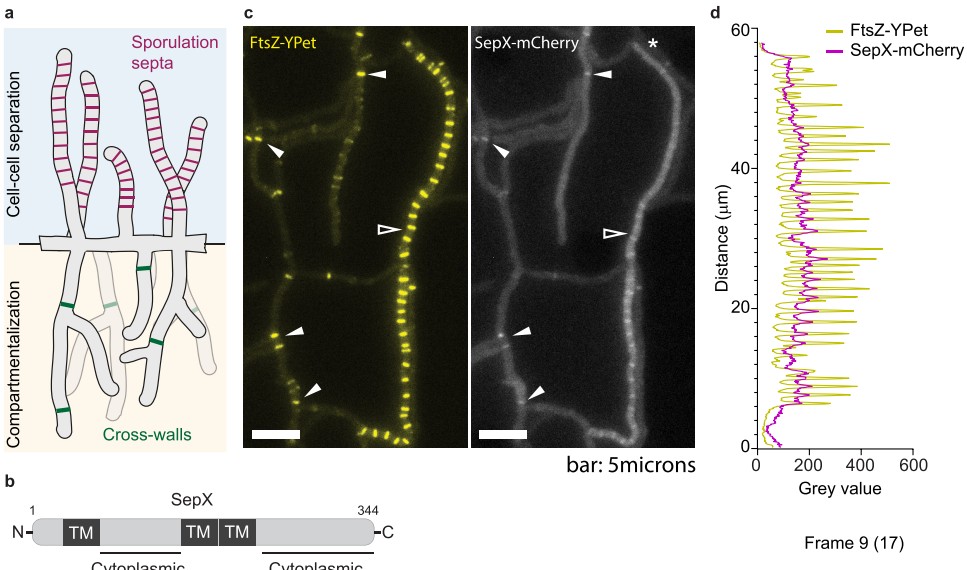

**Fig. 1 SepX co-localizes with FtsZ at cross-walls and sporulation septa. a** Schematic showing the two kinds of division septa that are formed in *Streptomyces*: vegetative cross-walls (green) and sporulation septa (magenta). **b** Schematic depicting the predicted domain organization of SepX (vnz_14865), including three transmembrane (TM) segments and two cytoplasmic loops. Numbers denote amino acid count. **c** Microscopic analysis of a merodiploid strain expressing both a SepX-mCherry fusion and a FtsZ-YPet fusion (MB256) reveals that SepX and FtsZ co-localize at both cross-walls (filled arrows) and sporulation septa (open arrow). Shown is a single frame from Supplementary Movie 1. Experiments were performed in duplicate. Scale bar: 5 µm. **d** Intensity profiles of FtsZ-YPet and SepX-mCherry fluorescence extracted from the sporogenic hyphae (*) presented in panel (**c**). Source data are provided as a Source Data file.

As part of this study, we examined the localization of a protein encoded by *vnz_14865* (now named SepX for "septal protein X"), a target of WhiAB-mediated activation (Supplementary Fig. 1a). SepX is a putative 344-amino acid membrane protein that contains two largely unstructured regions predicted to reside in the cytoplasm (Fig. 1b). Further bioinformatic analysis indicated that SepX does not share any homology with proteins of known function. To visualize its subcellular localization, we constructed a merodiploid strain in which *sepX* was fused to *mcherry* and expressed in trans from its native promoter from the ΦBT1 integration site. This strain additionally carried a $P_{ftsZ}$-*ftsZ-ypet* gene fusion, thereby allowing us to fluorescently label both vegetative and sporulation septa. Microscopic analysis of this dual-labelled strain by time-lapse microscopy revealed that SepX-mCherry frequently co-localizes with FtsZ at both vegetative cross-walls and sporulation septa (Fig. 1c, d, Supplementary Fig. 1b and Supplementary Movie 1). Control experiments using single-labelled strains confirmed that the detected fluorescent signals for FtsZ-YPet and SepX-mCherry were specific (Supplementary Fig. 1c). We were unable to resolve whether SepX localization precedes the arrival of FtsZ, due to the weak fluorescent signal of the SepX-mCherry fusion when produced from its native promoter. The presence of SepX at both cell division sites is in line with transcriptomics data that showed expression of the *sepX* gene both during vegetative and reproductive growth. Furthermore, albeit at reduced levels, *sepX* transcripts were also detected in the absence of WhiA or WhiB (Supplementary Fig. 1d), indicating that expression of *sepX* might be controlled by an additional factor.

To determine if SepX accumulation at nascent division sites was dependent on FtsZ, we inserted the *sepX-mcherry* fusion into the Δ*ftsZ* null mutant in trans. SepX-mCherry was produced from the constitutive *ermE\** promoter to ensure expression in the mutant background. Subsequently, we inspected hyphae of the resulting strain by fluorescence microscopy. In addition, we used the fluorescent D-amino acid analogue HADA to label cell wall peptidoglycan, including cross-walls and sporulation septa[16]. As expected, *ftsZ*-deficient hyphae were devoid of division septa. In these continuous hyphae, SepX-mCherry was stably produced and appeared to be distributed evenly within the hyphae, indicating that the distinct localization pattern of SepX-mCherry observed in the wild type requires FtsZ (Supplementary Fig. 1e, f).

**SepX is required for compartmentalization and sporulation.** Intrigued by the accumulation of SepX at both types of division septa, we engineered an in-frame gene deletion and examined the importance of SepX for cell division during vegetative growth and sporulation. Using the fluorescent membrane dye FM4-64, we first visualized vegetative septa as a proxy for cross-walls in *S. venezuelae*, comparing the wild type to the Δ*sepX* mutant. Strikingly, deletion of *sepX* led to a vegetative mycelium almost completely devoid of cross-walls (Fig. 2a, b). To better understand the effect of a *sepX* deletion on the overall frequency of cross-wall formation, we quantified the number of cross-walls produced in hyphae during vegetative growth. To achieve this, we allowed spores of the wild type, the Δ*sepX* mutant and the complemented mutant (Δ*sepX/sepX*+) to germinate and grow in the presence of HADA in a microfluidic device for 5–6 h. We then imaged emerging hyphae by fluorescence microscopy and quantified the number of cross-walls per hypha. We found that in wild-type *S. venezuelae* cross-walls are synthesized on average every 20–30 μm (Fig. 2c). In agreement with the FM4-64 membrane staining (Fig. 2b), deletion of *sepX* led to an almost total lack of cross-walls within the vegetative mycelium. Very occasionally, cross-wall

formation occurred close to the "mother" spore, but even where such cross-walls were observed, the resulting vegetative mycelium was devoid of septa (Supplementary Fig. 2a). The frequency of cross-wall formation was restored when Δ*sepX* was complemented in trans.

Since a Δ*sepX* mutant lacks cross-walls, we wondered whether constitutive expression of *sepX* (*sepX*++) would lead to a converse increase in cross-wall frequency during vegetative growth. Therefore, we repeated our HADA labelling experiments with a strain that expressed *sepX* in trans from the constitutive *ermE\** promoter[17] (Supplementary Fig. 2e). While we did not observe any visible effects on growth and sporulation of *S. venezuelae* (Supplementary Fig. 2b), we found that approximately double the number of cross-walls could be observed in the SepX++ strain compared to the wild type, with a cross-wall forming every 10–15 μm (Supplementary Fig. 2c, d). Constitutive expression of other divisome components that are also part of the division machinery involved in vegetative cell division and cross-wall formation, including *sepF* and *ftsZ* itself, did not lead to an increase in cross-wall frequency (Supplementary Fig. 2c, d). Thus, these results indicate that SepX abundance is a determinant for FtsZ-dependent vegetative cell division leading to hyphal compartmentalization.

Notably, our membrane staining experiments also showed that following vegetative growth, *sepX*-deficient hyphae did initiate sporulation-specific septation (Supplementary Fig. 2f, g). However, sporulation septa that formed under these conditions were irregularly spaced or often appeared tilted, compared to the evenly spaced septa that were visible in wild-type hyphae. To further investigate the importance of SepX for sporulation-specific cell division, we imaged sporulating hyphae of the wild type and the Δ*sepX* mutant by Transmission Electron Microscopy (TEM) and cryo-Scanning Electron Microscopy (SEM). Inspection of electron micrographs of aerial hyphae revealed that, in contrast to the wild type and the complemented mutant, little to no sporulation was observed after 3 days in *sepX*-deficient hyphae. Given more time (+4 days), colonies of the Δ*sepX* mutant did sporulate, but hyphae failed to deposit regularly placed septa, resulting in spores of variable length (Supplementary Fig. 2h–j). In line with the observed spore morphology, we found that spores produced by Δ*sepX* mutant were consistently larger and more variable in size compared to spores from the wild type and the complemented mutant (Fig. 2d–e). Partial septation from one side of the sporogenic hypha was also frequently observed, suggesting FtsZ-mediated constriction had been initiated but then stalled or aborted (Fig. 2f). Most strikingly, septation occasionally appeared to occur in an altered plane, often perpendicular to the normal plane of cell division. Taken together our results suggest that SepX plays a crucial role in cell division during vegetative growth and sporulation in *Streptomyces*.

**Lack of hyphal compartmentalization is crucial for fitness.** In addition to the severe cell division defects we observed by electron and light microscopy, colonies formed by the Δ*sepX* mutant also displayed a striking macroscopic phenotype. Wild-type *S. venezuelae* forms roughly circular colonies, that are several millimetres in diameter after three days and produce the green polyketide pigment characteristic of mature *S. venezuelae* spores (Fig. 3a). In contrast, the Δ*sepX* mutant grew much more slowly, producing colonies that were small and aberrant in shape (Fig. 3b). Colonies remained white, characteristic of a sporulation defect, as has been described for other developmental mutants[18]. This growth phenotype could be complemented by expressing *sepX* from the native promoter in trans (Fig. 3c). The Δ*sepX*

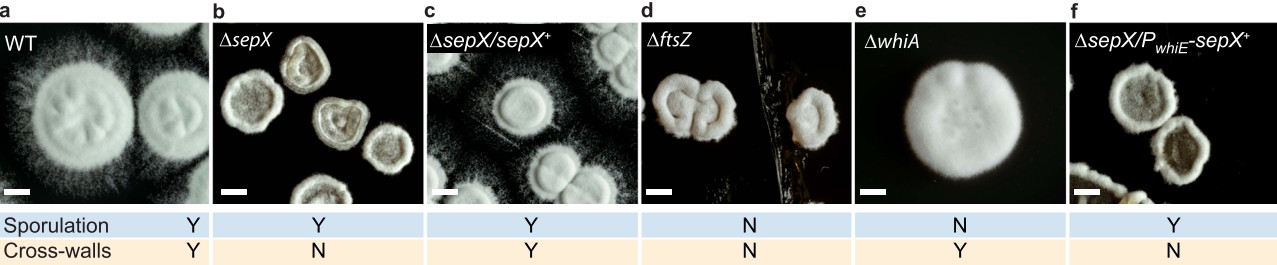

**Fig. 2 SepX is required for vegetative cross-wall formation and regular sporulation septation. a** Vegetative septa can be visualized by fluorescence microscopy after incubating wild-type (WT) *S. venezuelae* hyphae with the membrane dye FM4-64. **b** The *sepX* deletion strain (SV55) lacks septa in growing hyphae, as visualized by FM4-64 staining. Scale bars in **a** and **b**: 10 μm. Data shown in **a** and **b** are representatives of experiments performed in triplicate. **c** Quantification of cross-wall frequency in hyphae emerging from spores of the wild type (WT), the Δ*sepX* mutant (SV55) and the complemented mutant Δ*sepX/sepX*⁺ (MB745). Cross-walls were visualized by growing the strains in the presence of 0.25 mM HADA for 5–6 h following germination. Data were derived from triplicate experiments and at least 20 spores per replicate were analyzed. Solid lines represent simple linear regression. **d** Spore volume frequency distributions (normalized to total count, *n* = 100,000) of the wild type (WT), the Δ*sepX* mutant (SV55) and the complemented mutant (MB181). Shown is the mean ± s.e.m. of three biological replicates per strain. **e** Representative phase contrast images of spores analyzed in (**d**). Data shown are representatives of experiments performed in triplicate. Scale bars: 5 μm. **f** TEM images of sporulating hyphae in the wild type (WT), the Δ*sepX* mutant (SV55) and the complemented mutant Δ*sepX/sepX*⁺ (MB181). Scale bars: 500 nm. Shown are representative images of at least 5 independent hyphae. Source data are provided as a Source Data file for (**c**) and (**d**).

**Fig. 3 Cross-walls are crucial for cellular development of *S. venezuelae*.** Representative images of the colony morphology of **a** the wild type (WT), **b** the Δ*sepX* mutant (SV55), **c** the complemented mutant (MB181), **d** the Δ*ftsZ* mutant (DU669), **e** the Δ*whiA* mutant (SV7), or **f** the conditional *sepX* mutant expressing *sepX* from the sporulation-specific *whiE* promoter (MB1120). All strains were grown on solid MYM and photographed after 3 days. "Y" (Yes) and "N" (No) indicate whether strains can form cross-walls and/or sporulation septa. Scale bars: 1 mm.

colony morphology is reminiscent of the phenotype of an Δ*ftsZ* mutant, which is devoid of cross-walls and sporulation septa (Fig. 3d). Notably, unlike the *ftsZ* and *sepX* deletion mutants, colonies of *Streptomyces* mutants that produce cross-walls but are specifically blocked in sporulation septation, such as the Δ*whiA* mutant[14] (Fig. 3e), grew to a similar size and morphology as the wild type. This implies that the inability to synthesize cross-walls and thus compartmentalize the growing mycelium is associated with a severe fitness penalty.

To further dissect the importance of *sepX*-dependent cross-wall formation for cellular development and sporulation, we complemented the Δ*sepX* mutant with a SepX-mCherry fusion that was specifically expressed during sporulation but not during vegetative growth, thereby impairing cross-wall formation but permitting sporulation-specific cell division. Based on transcriptional profiling data[19] (Supplementary Fig. 3a), we placed *sepX-mcherry* under the control of the sporulation-specific promoter upstream of the *whiE* operon, which directs the synthesis of the polyketide that leads to the pigmentation seen in mature *Streptomyces* spores[20]. Fluorescence microscopy and Western blot analysis confirmed that $P_{whiE}$ was effective in limiting SepX synthesis to the reproductive growth phase compared to strains in which *sepX-mcherry* expression was driven either from the constitutive *ermE*\* or the native promoter (Supplementary Fig. 3b, c). As expected, the absence of SepX-mCherry during vegetative growth led to an almost complete lack of cross-walls (Supplementary Fig. 3d). This conditional *sepX* mutant was noticeably similar in morphology to the Δ*sepX* mutant (Fig. 3f). However, sporulation-specific septation occurred readily in this strain to generate spores that were largely regular in size and shape, similar to the Δ*sepX*/$P_{sepX}$-*sepX-mcherry* complementation strain, suggesting that SepX was functional during sporulation (Supplementary Fig. 3e, f). Collectively, these data suggest that SepX-mediated hyphal compartmentalization but not sporulation is crucial for growth and cellular development in *Streptomyces*.

**SepX promotes stable Z-ring formation during sporulation**. Next, to further determine the role of SepX in sporulation-specific cell division, we conducted time-lapse fluorescence microscopy, tracking the assembly of FtsZ-YPet into Z-rings in wild-type *S. venezuelae* and the Δ*sepX* mutant. In the wild type, following the cessation of hyphal growth, Z-ladder formation occurred rapidly and near synchronously within the tip compartment of a hypha (Supplementary Movie 2). In contrast to the wild type and in line with the lack of hyphal compartmentalization, we observed frequent cell lysis and highly abundant Z-ladder formation throughout the entire mycelium of the *sepX*-deficient strain (Fig. 4a, b and Supplementary Movies 2 and 3). While the assembly of Z-ladders in the wild type led to regular septation and the generation of chains of spores, timely completion of the sporulation process in the Δ*sepX* mutant appeared much less efficient. This was further confirmed by quantifying the number of spores produced by the wild type, the Δ*sepX* mutant and the complemented mutant (Supplementary Fig. 4c). The visible septation defects in the Δ*sepX* mutant at the single-cell level are in agreement with the electron micrographs of Δ*sepX* spore chains, showing the absence of expected septa or incomplete septation (Fig. 2f and Supplementary Fig. 2h, i). Furthermore, kymographs of FtsZ-YPet fluorescence from sporulating wild-type and Δ*sepX* hyphae, suggested a clear difference in the dynamics of Z-ring formation and stability (Fig. 4c, d). Analysis of individual Z-rings confirmed the time-dependent and localized increase in FtsZ-YPet fluorescence intensity in wild-type hyphae, indicating the maturation and constriction of Z-rings before they eventually disassembled after ~2 h (Fig. 4e). In *sepX*-deficient hyphae,

however, Z-rings assembled much more slowly and displayed a lower average fluorescence intensity. Furthermore, although the spacing of Z-rings was similar to the wild type (Fig. 4f), individual Z-rings in Δ*sepX* hyphae were on average spatially less confined compared to wild-type hyphae (Fig. 4g). Notably, Western blot analysis confirmed that the alteration in Z-ring dynamics observed in the absence of *sepX* was not due to a reduction in the levels of FtsZ. We found that the abundance of FtsZ, the sporulation-specific cell division protein SsgB[8] and the developmental regulator WhiA[14] was increased compared to the wild type (Supplementary Fig. 4f). This suggests that the absence of *sepX*, and consequently the lack of cross-walls, affects the abundance of both structural and regulatory proteins involved in cell division, which directly or indirectly influences the correct assembly and architecture of fission-competent Z-rings.

**SepX interacts with the cell division machinery**. SepX is comprised of three predicted transmembrane segments connected by two largely disordered and cytoplasmic regions (Fig. 1b). Given the lack of any obvious catalytic domain, we reasoned that SepX might fulfil a structural role to support the assembly of the cell division machinery. To test this idea, we performed a targeted bacterial two-hybrid analysis, screening for interaction between SepX and known *Streptomyces* divisome components, including SepF, the two additional SepF-like proteins SepF2 and SepF3, the two dynamin-like proteins DynA and DynB and the actinomycete-specific cell division protein SsgB[8,11]. We were unable to test for an interaction with FtsZ because we previously found that *S. venezuelae* FtsZ is not functional in this assay[11]. Using β-galactosidase assays to assess the interaction between the individual two-hybrid combinations, we found that SepX self-interacted and interacted with DynB but did not bind the DynB partner protein DynA or any of the other divisome components tested (Fig. 5a and Supplementary Fig. 5a). Like SepX, DynB is also associated with the membrane via two transmembrane segments[11]. Our two-hybrid results further suggest that the DynB membrane domains may be involved in SepX binding because a DynB mutant version lacking the transmembrane segments failed to bind SepX (Supplementary Fig. 5a). To verify that the observed interaction was specific, we also included the transmembrane anti-sigma factor RsbN, which strongly interacts with its cognate sigma factor, σ[BldN] [19], in our two-hybrid analyses. Importantly, neither SepX nor DynB interacted with RsbN, supporting the notion that SepX specifically associates with DynB (Supplementary Fig. 5b).

To further corroborate the two-hybrid results, we performed co-immunoprecipitation coupled to mass spectrometry with a Δ*sepX* strain that produced a FLAG-tagged copy of *sepX* from its native promoter in trans. Sporulating cultures of this strain were cross-linked and SepX-FLAG was recovered from the cell lysate using anti-FLAG antibody-conjugated magnetic beads. Proteins retained on the beads were eluted and analyzed by mass spectrometry. In parallel, we performed control experiments with an untagged wild-type strain to identify proteins that bound non-specifically to the anti-FLAG antibody or to the beads. Inspection of obtained peptide profiles revealed a clear enrichment of SepX in the FLAG samples compared to the control samples. Other highly represented proteins in our analysis included metabolic enzymes and transporters. It is conceivable that these proteins are very abundant in the cell and reside in close proximity to SepX, resulting in the observed enrichment of these proteins in our analysis. Importantly, we detected at least 2-fold enrichment of DynB and its partner protein DynA, and several additional cell division proteins, including SepF, SepF2, SsgB and FtsK (Fig. 5b). Previous work had demonstrated that

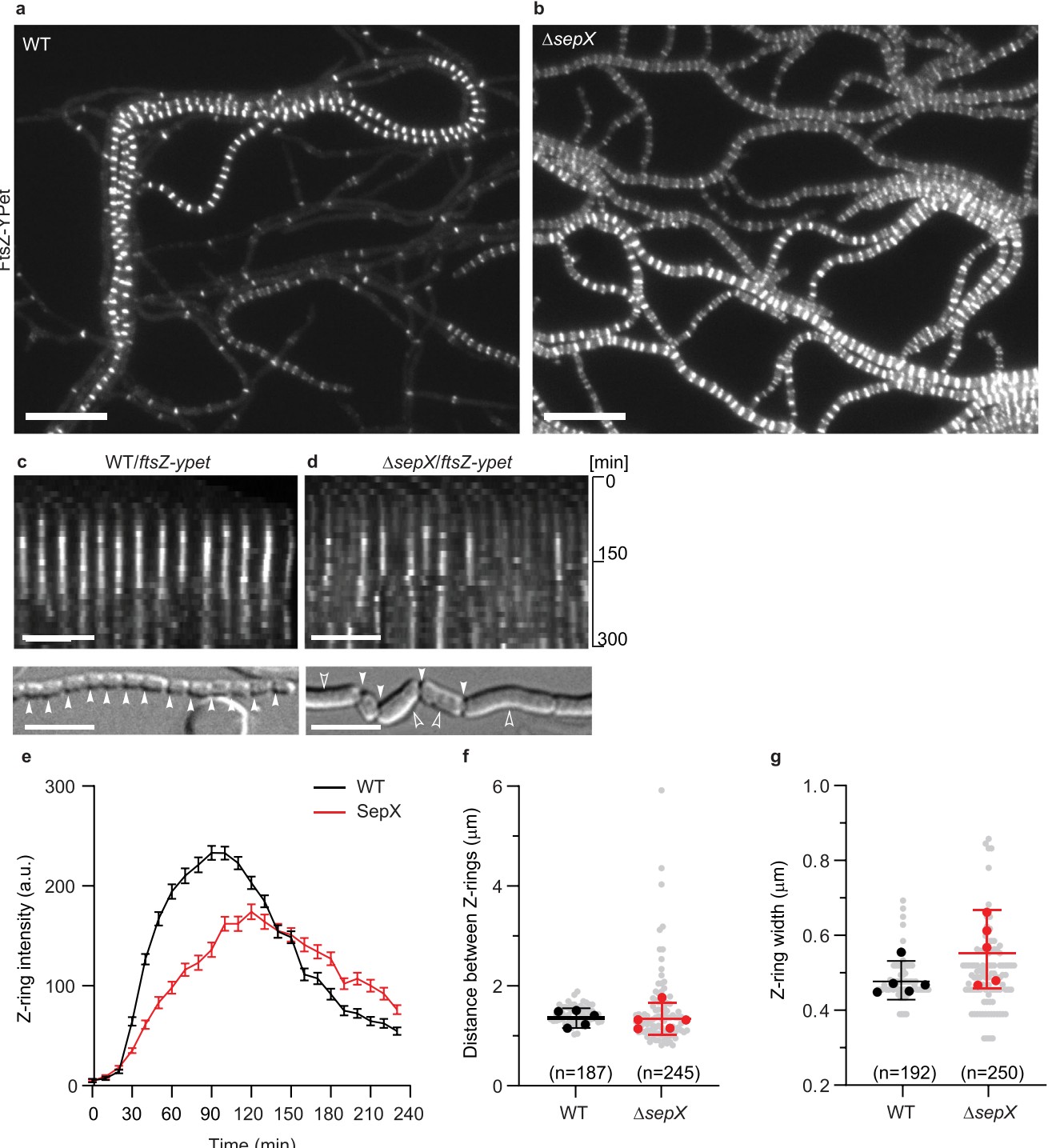

**Fig. 4 SepX is important for Z-ring stability and architecture during sporulation-specific cell division.** Still images from Supplementary Movie 2 and 3 showing YPet-labelled Z-ring formation in sporulating hyphae of **a** the wild type (SS12) or **b** the Δ*sepX* mutant (MB180). Scale bar: 10 μm. **c** and **d** Subsection of representative kymographs (top panel) showing the spatiotemporal localization of FtsZ-YPet in **c** the wild type (WT, SS12) and **d** the Δ*sepX* mutant (MB180). *Y*-axis shows imaging duration. The bottom panel depicts terminal sporulation phenotype with filled arrowheads pointing at completed sporulation septa, open arrowheads at failed septation. Scale bars: 5 μm. Complete kymographs are shown in Supplementary Fig. 4d, e. **e** Fluorescence intensity traces of FtsZ-YPet (Z-rings) over time derived from sporulating WT (SS12) and Δ*sepX* mutant hyphae (MB180). Shown are the mean fluorescence intensity traces (mean ± s.e.m.) collected from Z-rings of five sporulating hyphae for each strain (*n* = 5 independent hyphae per strain). **f** Distance and **g** width of Z-rings in sporulating hyphae of WT (SS12) and *sepX*-deficient hyphae (MB180). The analysis is based on the same data time-lapse data that was used to generate panel (**e**). Shown are the mean values for Z-ring distance and width for reach replicate (black and red dots) ± 95% CI. *n*-number of Z-rings from five representative hyphae per strain were analyzed. The data shown here are representative of experiments performed in duplicate. Source data are provided as a Source Data file for (**e**–**g**).

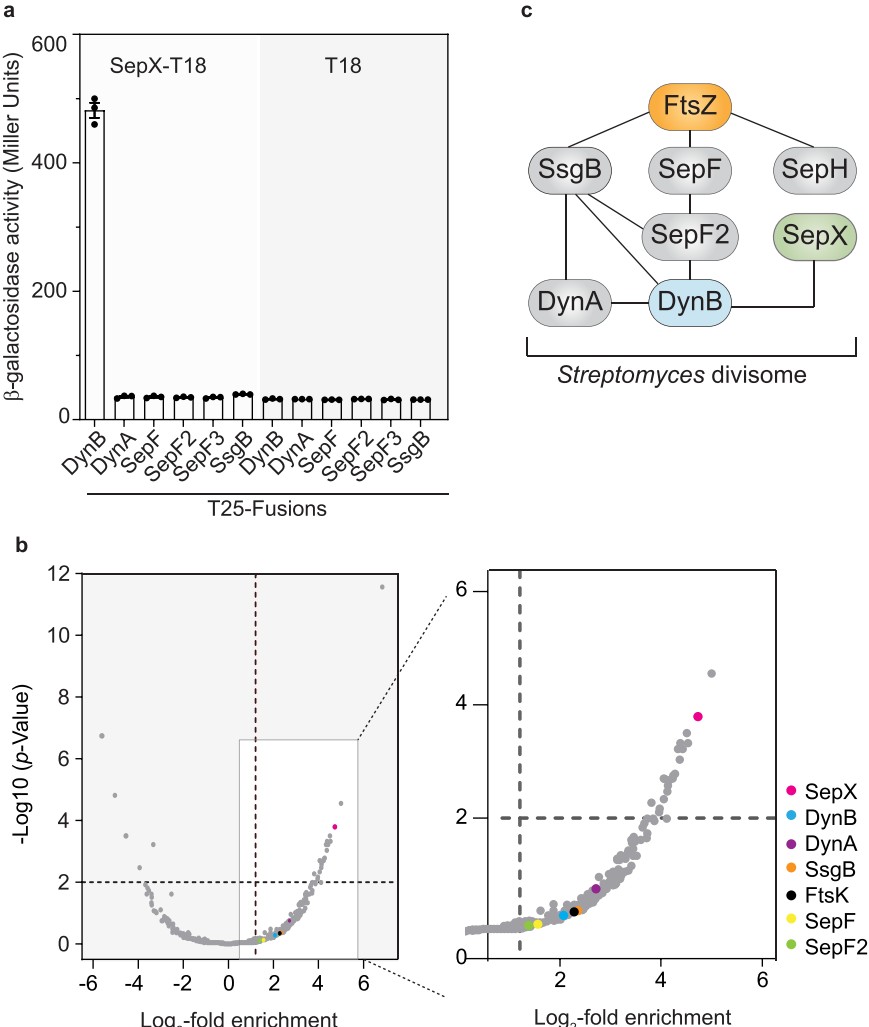

**Fig. 5 SepX is part of the _S. venezuelae_ divisome and specifically interacts with DynB. a** Bacterial two-hybrid analysis and corresponding β-galactosidase activities between SepX-T18 or the empty plasmid (T18) and the indicated T25-fusions. Data points present mean values ± s.e.m. Each interaction was assayed in biological triplicate. **b** Co-immunoprecipitation and mass spectrometry analysis of SepX. Immunoprecipitation experiments were performed using the control strain SS92 (_dynB-ypet_) or a strain expressing _a sepX-FLAG/dynB-ypet_ fusion (MB942) and anti-Flag antibody-conjugated magnetic beads. Proteins that were retained on the beads were analyzed using mass spectrometry. Data obtained from three biological replicate experiments are displayed as a volcano plot (left) with several putative interaction partners highlighted in the enlarged subsection of the plot (right). The complete data set is included in the Source Data. **c** Schematic showing the proposed SepX interaction network, including DynB and several known _S. venezuelae_ divisome components[11,12].

DynB is involved in Z-ring stabilization during sporulation-specific cell division and associates with the divisome by binding to SsgB and SepF2, which in turn are either directly (SsgB) or indirectly (SepF2) connected with the Z-ring[8,11]. Thus, the enrichment of other key divisome components in our analysis could have been mediated by the SepX–DynB interaction. In summary, our protein-protein interaction studies place SepX in a network of interactions involving key divisome components involved in FtsZ-ring positioning, anchoring and stability (Fig. 5c).

**Sporulation requires SepX or the dynamin-like protein DynB.** We noted that the sporulating hyphae of the Δ_sepX_ mutant often displayed asymmetric and incomplete constrictions (Fig. 2f). This phenotype had previously also been reported for _an S. venezuelae_ strain lacking the two divisome-associated proteins DynA and DynB, which are both required for wild type-like sporulation[11]. Given that the Δ_sepX_ and the Δ_dynAB_ mutant share a similar sporulation defect, that all three proteins co-localize with FtsZ at

sporulation septa and that SepX and DynB physically interact in vivo, we asked if SepX and DynB are functionally redundant or fulfil an independent role during sporulation-specific cell division.

To address this question, we first tested whether the interaction between SepX and DynB is required for the accumulation of either protein at nascent sporulation septa. Fluorescent microscopy revealed that SepX-mCherry was able to localize to both vegetative cross-walls and sporulation septa in the absence of _dynAB_ (Fig. 6a). Likewise, YPet-labelled DynB accumulated in the expected ladder-like pattern during sporulation in the absence of _sepX_ (Fig. 6b). Therefore, while SepX and DynB may interact, this interaction is not required for the recruitment of either component to future division sites. Furthermore, expression of _sepX_ from the constitutive _ermE*_ promoter could not suppress the sporulation defects observed in the Δ_dynB_ mutant. Equally, expression of _dynAB_ from the _ermE*_ promoter could not complement the Δ_sepX_ mutant (Supplementary Fig. 6a–e, h and i). These findings suggest that DynB and SepX function independently during sporulation-specific cell division.

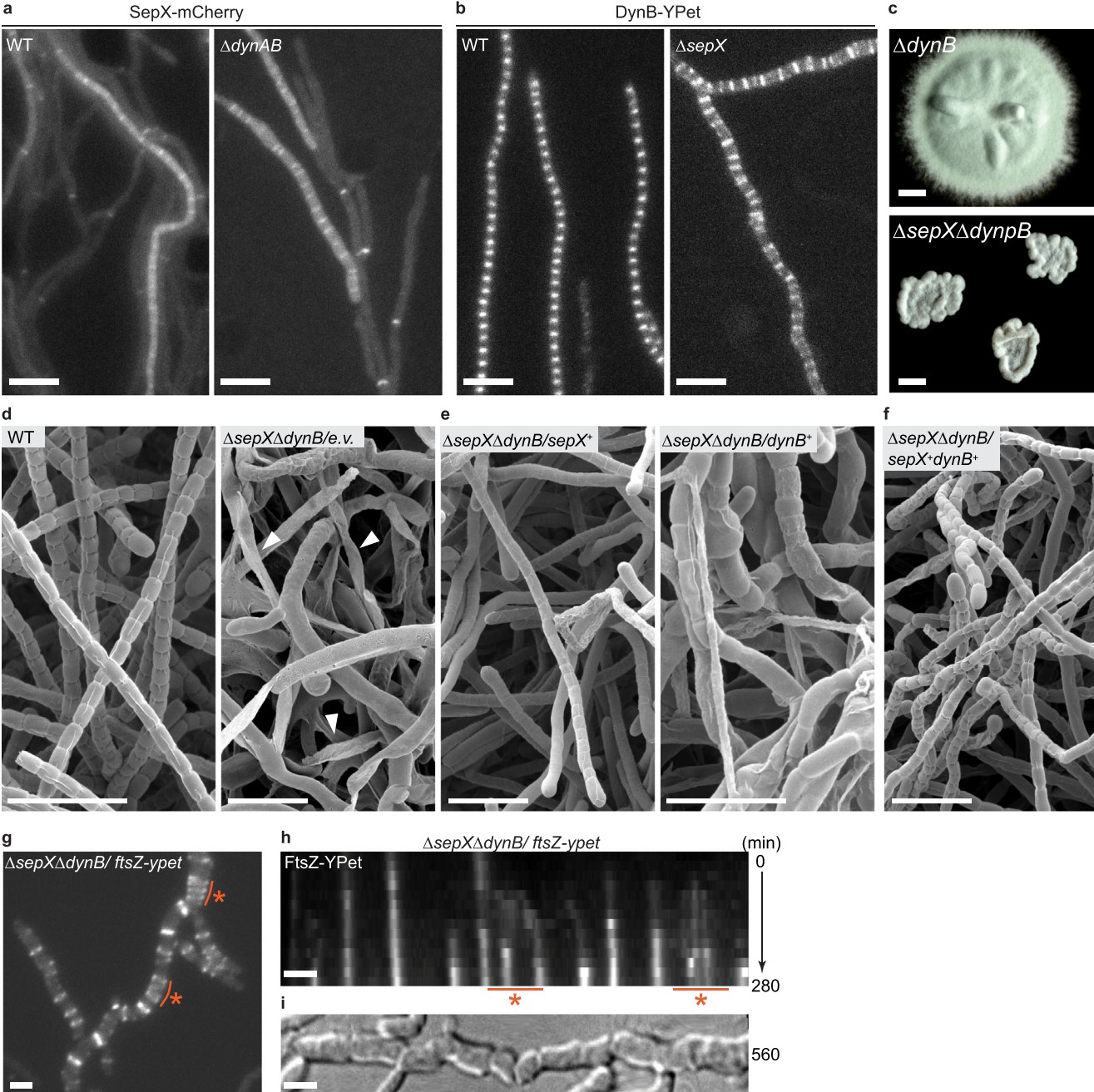

**Fig. 6 SepX and DynB function independently during sporulation but cause a synthetic sporulation defect when deleted together. a, b** Microscopic analysis showing the subcellular localization of SepX-mCherry or DynAB-YPet in the wild type (MB256, SS92) or in the absence of either *dynAB* (MB1270) or *sepX* (SS155). Data shown are representatives of experiments performed in duplicate. Scale bars: 5 μm. **c** Colony morphology of the Δ*dynB* mutant (SS2) compared to the Δ*sepX*Δ*dynB* double mutant (SV57). Strains were grown for 3 days on solid MYM. Scale bar: 1 mm. (**d–f**) Cryo-SEM images of *S. venezuelae* sporogenic hyphae, including (**d**) the wild type (WT) and the Δ*sepX*Δ*dynB* double mutant carrying an empty plasmid (MB1099), white arrow heads in right panel point at lysed hyphae, **e** the Δ*sepX*Δ*dynB* double mutant complemented with either *sepX* (MB1102) or *dynB* (MB1101) expressed from the *ermE** promoter (+), or **f** the Δ*sepX*Δ*dynB* double mutant complemented with both *sepX* and *dynB* (MB1103). Scale bars: 5 μm. Data shown in **d–f** are representatives of at least seven images. **g** Still images taken from Supplementary Movie 3 showing FtsZ-YPet localization in the Δ*sepX*Δ*dynB* mutant (MB1111). Orange line and asterisk denote regions of migrating Z-rings. Scale bar: 2 μm. **h, i** Kymograph showing FtsZ-YPet distribution in a sporulation Δ*sepX*Δ*dynB* hypha presented in **g** with the corresponding DIC image (**i**) displaying the terminal sporulation phenotype. Orange line and asterisk in (**h**) correspond to the region highlighted in **g**. Scale bars: 2 μm. Data shown in **g–i** are representatives for experiments performed in duplicate.

Since SepX and DynB are both individually important for normal sporulation in *S. venezuelae*, we additionally engineered a strain of *S. venezuelae* that lacked both *sepX* and *dynB*. Unlike the parental Δ*dynB* mutant, colonies of the Δ*sepX*Δ*dynB* double mutant were small, reflecting the fitness cost we had shown to be associated with the deletion of *sepX* and the resulting absence of vegetative cross-walls (Fig. 6c). In contrast to Δ*dynB* and Δ*sepX* single mutants, which sporulate, albeit aberrantly (Supplementary Fig. 6a–c), cryo-SEM analysis showed that the Δ*sepX*Δ*dynB* double mutant largely failed to sporulate on solid media and

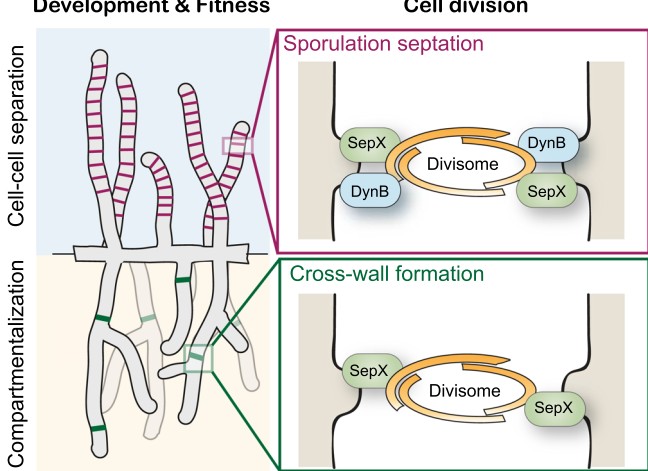

**Fig. 7 Proposed model for SepX function in *Streptomyces* cell division.** During vegetative growth, SepX is crucial for the establishment of stable Z-rings leading to cross-wall formation and the compartmentalization of the hyphae which in turn is required for the coordinated progression through the life cycle and overall fitness of the organism. During reproductive growth, SepX functions together with the dynamin-like protein DynB to ensure the spatial confinement of Z-rings, thereby stabilizing divisome assembly, function and efficient sporulation septation.

displayed extensive lysis frequently within the mycelium (Fig. 6d). Occasionally we observed a very low number of spore-like cells at the outer edge of the colony, which were highly aberrant in size and shape (Supplementary Fig. 6f). Complementation of the double mutant with *dynB* reconstituted the Δ*sepX* mutant phenotype while complementation with *sepX* reconstituted the Δ*dynB* mutant phenotype, resulting in irregular sporulation in both cases (Fig. 6e). Wild type-like sporulation could only be restored when the double mutant was complemented with both *sepX* and *dynB* (Fig. 6f).

In addition, we examined the localization of FtsZ-YPet in the Δ*sepX*Δ*dynB* double mutant using time-lapse fluorescence microscopy. During vegetative growth, FtsZ-YPet displayed a similar distribution compared to the Δ*sepX* mutant, with FtsZ-ring-like structures that sporadically assembled and continued to migrate in growing hyphae (Supplementary Movie 4). Consistent with the severe growth and septation defects of the Δ*sepX*Δ*dynB* double mutant on solid medium (Fig. 6c), we also found that the mutant hyphae were prone to frequent large-scale lysis events. In the few hyphae where sporulation septation was completed, FtsZ-YPet localization was highly aberrant, with the Z-rings being irregularly laid down and much less spatially confined compared to the Z-rings established in the wild type (Fig. 4c and Fig. 6g–i). As observed for the Δ*sepX* mutant strain, FtsZ levels in the Δ*sepX*Δ*dynB* double mutant were similar to those of the wild type and the Δ*dynB* mutant (Supplementary Fig. 6g), indicating the phenotype arises from a defect in the efficient assembly of division-competent Z-rings, rather than insufficient FtsZ.

Finally, given the complexity of the *Streptomyces* life cycle, we performed phylogenetic analysis to determine whether SepX and DynB are predominantly found in *Streptomyces* species, or are also present in other actinobacterial systems that divide by binary fission. We first searched 62-representative Streptomycetaceae genomes and confirmed the co-conservation of SepX and DynB homologues throughout the genus (Supplementary Fig. 7). In support of the conserved distribution of *sepX*, we found that expressing the *sepX* homologue from *Streptomyces coelicolor* in *S. venezuelae* can complement the Δ*sepX* mutant phenotype and restore normal growth and sporulation (Supplementary Fig. 8).

This clearly indicates that SepX function is conserved among *Streptomyces* species. In the wider actinobacteria, a search of 673 representative genomes revealed that *sepX-dynB* co-occurrence is also frequently found outside of the genus *Streptomyces*, including in unicellular actinobacteria (Supplementary Fig. 9). Although SepX homologues share <10% identical residues, visualization of multiple sequence alignments revealed that residues predicted to reside in the membrane are more conserved (Supplementary Fig. 10a and b). These results suggest that the association of SepX with the cytoplasmic membrane could be crucial for its function and interaction with DynB during cell division.

Taken together, our results demonstrate that SepX and DynB have distinct but synergistic roles in ensuring the stability and function of the Z-ring in *Streptomyces*. In addition, sporulation-specific cell division cannot occur effectively in the absence of both SepX and DynB, further underlining their collective importance as central components of the *Streptomyces* cell division machinery.

## Discussion

The *Streptomyces* life cycle is characterized by two distinct FtsZ-dependent modes of cell division that lead to the separation of growing hyphal filaments into multigenomic compartments and to the formation and release of unigenomic spores. While some recent progress has been made towards the understanding of the components involved in sporulation-specific sporulation[6,8,11,12], little has been known about the determinants for FtsZ-mediated cross-wall formation. Here we describe the previously unknown cell division protein SepX, which is crucial for cross-wall formation and sporulation in *Streptomyces*. Our results suggest that SepX functions as a structural component of the divisome and contributes to the spatial confinement and stability of the divisome during vegetative growth and sporulation (Fig. 7).

Our work reveals that SepX accumulates with FtsZ at future division sites in both vegetative and sporulating hyphae (Fig. 1). We note that SepX-mCherry localization appears to be more stable at vegetative septa compared to sporulation septa, implying that in sporulating hyphae SepX localization is more flexible due to the presence of DynB and possibly other sporulation-specific cell division factors. Deletion of *sepX* causes severe cell division defects during both vegetative and reproductive growth, leading to the lack of hyphal compartmentalization and aberrant sporulation (Fig. 2). Importantly, apart from FtsZ, which is essential to form both vegetative and sporulation septa[6,21], we identified SepX as the first protein specifically required for cross-wall formation.

We show that the inability to divide the mycelial network into compartments results in a fitness cost, reduced growth, and aberrant colony morphology (Fig. 3). In agreement with previous work using the *ftsZ* null mutant[22], we found that in the absence of cross-walls, growing hyphae were prone to lysis. This is in line with the idea that cross-walls protect the mycelial mass from large-scale loss of viability[22]. Furthermore, we propose that the organization of hyphae into compartments could also promote the ordered progression through the *Streptomyces* life cycle. For example, unlike the wild type in which Z-rings assemble in the tip compartment of sporogenic hyphae, there is a striking sudden appearance of Z-rings, which spread through the entire mycelium in *sepX*-deficient hyphae (Fig. 4a and b, Supplementary Movie 2 and 3). In line with these results, we found that the levels of FtsZ, the Z-ring binding protein SsgB and the transcriptional regulator WhiA were increased in *sepX*-deficient hyphae compared to the wild type (Supplementary Fig. 4f). However, sporulation appeared to be significantly delayed and less efficient in the absence of *sepX* (Supplementary Fig. 4c). In support of these results, recent work

by Zacharia et al. (2021)[23] revealed that the spatial organization of specialized cell types within *S. coelicolor* colonies is determined by the ordered spatiotemporal expression of key developmental regulators. Although the exact molecular mechanisms are still unclear, it is conceivable that hyphal compartmentalization during vegetative growth is required for the compartment-specific accumulation of developmentally regulated proteins or signalling molecules to allow ordered progression through the life cycle.

We further demonstrate that SepX plays a direct role in the assembly process of Z-rings during sporulation. In the wild type, FtsZ-YPet accumulates as tight, condensed bands that display a time-dependent increase and decrease in fluorescence intensity, which correlates with the assembly, maturation and constriction of Z-rings. However, in *sepX*-deficient hyphae, Z-rings displayed a lower fluorescence intensity throughout the division process and were on average 10% wider, indicating that Z-rings are spatially less confined. In support of this, we also note that sporulation septa that formed under these conditions were often placed ectopically and divided spores diagonally or perpendicular to the division plane (Fig. 2f, g). A similar observation was recently reported for several point mutants in the early *Streptomyces* cell division protein SsgB[24], suggesting that, like SsgB, SepX plays an important role during the initial stages of divisome formation.

We found that SepX co-purifies with several known *Streptomyces* divisome components and binds the sporulation-specific divisome component DynB (Fig. 5). DynB stabilizes Z-rings via interaction with several divisome components, including DynA, SepF2 and SsgB and likely facilitates the association of SepX with this protein interaction network. Importantly, while SepX and DynB are partially dispensable for sporulation-specific cell division individually, a Δ*sepX*Δ*dynB* double mutant largely fails to produce viable spores (Fig. 6). In the rare cases where we did observe a sporulation-like process, individual Z-rings were often loose and displayed a dynamic behaviour, including spiral-like FtsZ-structures (Fig. 6g), demonstrating the combined importance of SepX and DynB for the spatial stabilization of the divisome and efficient sporulation septation.

How might SepX function in cell division? The *sepX* gene is not in the vicinity of genes encoding other known divisome components in *Streptomyces* or the wider actinomycetes (Supplementary Fig. 10c). SepX shows no homology to proteins with a known activity but is predicted to be membrane-bound and thus, could act as a membrane anchor for the division machinery. Although we did not detect a direct association between SepX and FtsZ, SepX could mediate its effect on Z-ring assembly and stability via a yet unidentified interaction partner. Further work is required to fully understand the mechanistic role of SepX during cell division and cellular differentiation.

## Methods

**Bacterial strains, oligonucleotides and growth media**. Strains, plasmids and oligonucleotides used in this work are listed in Supplementary Tables 1, 2 and 3 respectively. The *Escherichia coli* K-12 cloning strains TOP10 or DH5α were used to propagate plasmids and cosmids. BW25113[25] containing the λ RED plasmid, pIJ790, was used to create disrupted cosmids. When required, the following antibiotics were added to the growth medium: 100 μg/ml carbenicillin 50 μg/ml kanamycin, 25 μg/ml hygromycin, 50 μg/ml apramycin or 25 μg/ml chloramphenicol.

*S. venezuelae* was cultured in maltose-yeast extract-malt extract medium (MYM), supplemented with R2 trace element solution at 1:500. Liquid cultures were grown in flasks with springs at 30 °C at 250 rpm. When required, MYM agar contained 5 μg/ml kanamycin, 25 μg/ml hygromycin, 50 μg/ml apramycin or 50 μg/ml thiostrepton. Cosmids and plasmids were conjugated from the *E. coli* strain ET12567 containing pUZ8002[26] into *S. venezuelae* as described previously by Bush et al.[14].

**Construction and complementation of the Δ*sepX* mutant**. Using 'Redirect' PCR targeting[27], the Δ*sepX* mutant was generated in which the central (968 bp) coding region was replaced with a single apramycin resistance cassette. Cosmid 5E03

(http://strepdb.streptomyces.org.uk/) was introduced into *E. coli* BW25113 containing pIJ790 and the *sepX* gene (*vnz_14865*) was replaced with the *apr-oriT* cassette amplified from pIJ773 using the primer pairs mb266 and mb267. The resulting disrupted cosmids were confirmed by PCR analysis using the flanking primers mb1431 and mb1432, and introduced into *S. venezuelae* by conjugation via *E. coli* ET12567/pUZ8002. Homologous recombination gave rise to two distinct morphotypes of exconjugants and their genotypes were confirmed using the flanking primers mb1431 and mb1432. The faster-growing and larger exconjugants were all confirmed to be single cross-over (Apr^R, Kan^R) strains in which the wild-type gene was still present. All double cross-over strains (Apr^R, Kan^S) arose from smaller exconjugants that took 7–10 days to emerge. A representative *sepX* null mutant was designated SV55. For complementation, pMB182 was introduced into the Δ*sepX* mutant by conjugation.

**Construction and complementation of the Δ*sepX*Δ*dynB* double mutant**. To generate a Δ*sepXdynB* double mutant, the central (968 bp) coding region of *sepX* on the 5E03 cosmid was instead replaced with the *hyg-oriT* cassette amplified from pIJ10700 using the identical primer pair mb266 and mb267. Disrupted cosmids were confirmed by PCR using mb1431 and mb1432 and introduced into the *S. venezuelae dynB::apr* mutant (SS2) by conjugation. As for the Δ*sepX::apr* mutant (SV55), double cross-overs originated from slow-growing exconjugants (Hyg^R, Kan^S) and were confirmed via PCR analysis using mb1431 and mb1432. A representative Δ*sepX::hyg* Δ*dynB::apr* null mutant was designated SV57. For complementation, pIJ10257 and the pIJ10257-derived plasmids pSS64 and pMB156 were introduced into the SV57 mutant by conjugation, serving to complement in trans either *sepX*, *dynB* or both *sepX* and *dynB*.

**Analysis of spore morphology and sporulation efficiency**. For analysis of spore size distributions, triplicate spore preparations were made in 3 ml of 20% glycerol and suspended in Isoton II (Beckman Coulter, Brea, CA, USA). The particle volume was then measured for a minimum of 100,000 particles per sample using a Multisizer 4e Coulter counter (Beckman Coulter) equipped with a 30 μm aperture tube (size range 0.6–18 μm, aperture current 600 μA). All measurements were conducted with logarithmic spacing into 400 bins using the Multisizer 4 software (Version 4.01) but are presented on a linear *x*-axis for clarity. For analysis of spore efficiency, triplicate spore preparations were measured in the same way but instead counting the total number of particles in 50 μl. The dilution factor (1:500 for all samples) was then used to calculate the number of particles per μl as presented.

**Cryo-scanning electron microscopy**. *Streptomyces* samples were mounted on an aluminium stub using Tissue Tek^R (BDH Laboratory Supplies, Poole, England) and plunged into liquid nitrogen slush. The sample was transferred onto the cryo-stage of an ALTO 2500 cryo-transfer system (Gatan, Oxford, England) attached to either a Zeiss Supra 55 VP FEG scanning electron microscope (Zeiss SMT, Germany) or an FEI Nova NanoSEM 450 (FEI, Eindhoven, The Netherlands). Sublimation of surface frost was performed at −95 °C for 3 min before sputter coating with platinum for 150 s at 10 mA. The sample was moved onto the main cryo-stage in the microscope for imaging at 3 kV and digital TIFF files were stored. All uncropped SEM images, including additional images of the same strain, are shown in Supplementary Figs. 11–15.

**Transmission electron microscopy**. *Streptomyces* colonies were excised from agar plates and fixed overnight in 2.5% (v/v) glutaraldehyde in 0.05 M sodium cacodylate, pH 7.3 and post-fixed in 1% (w/v) OsO4 in 0.05 M sodium cacodylate for 1 h at room temperature. After three, 15 min washes in distilled water, they were dehydrated through an ethanol series and infiltrated with LR White resin (London Resin Company, Reading, Berkshire) which was polymerized at 60 °C for 16 h. Using a Leica UC6 ultramicrotome (Leica, Milton Keynes), 90 nm sections were cut and stained with 2% (w/v) uranyl acetate and 1% (w/v) lead citrate before viewing at either 80 kV in a Jeol 1200 transmission electron microscope (Jeol UK, Welwyn Garden City) or at 200 kV in an FEI Tecnai 20 transmission electron microscope (FEI UK Ltd, Cambridge) and imaged using an AMT digital camera (Deben, Bury St Edmunds, UK) to record TIFF files.

**Light microscopy and image analysis**. For imaging protein localization in *S. venezuelae*, cells were grown in MYM medium for 10–20 h and a 2 μL sample of the culture was immobilized on a 1% agarose pad. *Streptomyces* hyphae or spores were visualized using a Zeiss Axio Observer Z.1 inverted epifluorescence microscope fitted with a sCMOS camera (Hamamatsu Orca FLASH 4), a Zeiss Colibri 7 LED light source, a Hamamatsu Orca Flash 4.0v3 sCMOS camera, and a temperature-controlled incubation chamber. Images were acquired using a Zeiss Plan Apochromat 100x/NA 1.4 Ph3 objective or a Zeiss Alpha Plan-Apo 100x/1.46 Oil DIC M27 objective with the following excitation/emission bandwidths: YFP (489–512 nm/520–550 nm), mCherry (577–603 nm/614–659 nm), CFP (401–445 nm/460–479 nm) and FM4-64 (577–603 nm/614–759 nm). Still images and time-lapse images series were collected using Zen Blue (Zeiss) and analyzed using Fiji[28].

Time-lapse fluorescence imaging of FtsZ-YPet and SepX-mCherry localization in *S. venezuelae* was performed as previously described[29]. Briefly, *S. venezuelae*

strains were grown in MYM medium for about 36 h at 30 °C and 250 rpm to reach complete sporulation. To isolate spores, mycelium was pelleted at $400 \times g$ for 1 min. Spores were loaded into B04A microfluidic plates (ONIX, CellASIC), allowed to germinate and grown by perfusing MYM for 3 h before medium was switched to spent-MYM medium. Spent-MYM was prepared from the 36-h sporulation culture by filtering the growth medium to remove spores and mycelia fragments using a 0.22 μm syringe filter. The media flow rate and temperature were maintained at 2 psi and 30 °C. Time-lapse imaging was started ~8 h after spores had germinated and images were acquired every 10–20 min, respectively, until sporulation was completed. For the detection of SepX-mCherry, imaging intervals were increased to 40 min.

For co-localization analysis, frames that showed a clear SepX-mCherry and FtsZ-YPet signal in sporulating hyphae were identified. Images were corrected for background fluorescence using the "subtracted_measured_background" plugin in Fiji[30] and the resulting fluorescence intensities for the selected hyphae were visualized using GraphPad Prism 9.

Kymograph and subsequent image analysis were performed as described in Ramos-Léon et al. (2021)[12]. Briefly, kymographs were generated from registered time-lapse image series of strain SS12 (WT/ftsZ-ypet) and SV55 (ΔsepX/ftsZ-ypet) using the Fiji Reslice plugin. Hyphae undergoing sporulation septation were first identified based on the characteristic FtsZ-YPet localization pattern following the cessation of tip extension. Twenty-four frames (10 min/frame) including one frame immediately before and 22 frames after the cessation of hyphal growth were isolated. Selected hyphae were "straightened" and a segmented line (5 pt) was manually drawn along the centre of the straightened hyphae. FtsZ-YPet fluorescence intensity was plotted along this line as a function of time (240 min) using the "Reslice" command. Kymographs were further annotated in Adobe Illustrator CS6. To visualize fluorescence intensities of Z-rings over time, time-lapse series were first corrected for background fluorescence by applying a custom Python script with a multi-Otsu thresholding algorithm[12]. The following steps were performed in Fiji: Z-rings were identified manually in time-lapse series and an ROI of $10 \times 20$ pixels was drawn around each Z-ring. The average fluorescence intensity values within each ROI were then collected and the mean fluorescence intensity trace of all Z-rings isolated from either wild-type or sepH-deficient hyphae was plotted using Graphpad Prism 9.

To determine the width and distance of Z-rings, an average fluorescence intensity projection for each of the time-series was first generated in Fiji. The corresponding fluorescence intensity trace along a segmented line (5 pt) manually drawn along the hyphal midline was then extracted, and the obtained data was further processed in R. For each strain, five independent time-lapse series were analysed. Peaks (which correspond to potential Z-rings) were identified using a custom R script and further filtered to remove false-positive peaks with a fluorescence intensity below 100. Z-ring width was calculated by measuring the full width at half maximum of the Z-ring peak in the fluorescence intensity profiles and Z-ring spacing was determined by measuring the distance between neighbouring peaks. Z-ring width and distance values were plotted using Graphpad Prism 9.

**FM 4-64 staining**. For membrane staining, *Streptomyces* hyphae were incubated with 0.5 mg/ml FM 4-64 Dye (N-(3-Triethylammoniumpropyl)24-(6-(4-(Diethylamino) Phenyl) Hexatrienyl) Pyridinium Dibromide) (Molecular Probes) for 15 min in the dark. Hyphae were then directly spotted onto a 1% agarose pad and visualized using a Zeiss Axio Observer Z1 microscope using an Alpha Plan-Apo 100×/1.46 Oil DIC M27 objective. Spores of SV55 were loaded into a BA04 microfluidic plate (ONIX, CellASIC) and allowed to germinate and to grow using constant perfusion of MYM containing 5.5 μg/ml FM4-64 at 30 °C. To promote sporulation, MYM/FM4-64 medium was replaced after 3 h of growth by "spent-MYM"/FM4-64. The "spent-MYM" was prepared as described previously[29]. Hyphae were visualized using a Zeiss Axio Observer Z1 microscope equipped with a Plan Apochromat 100×/1.4 Oil Ph3 objective. Images were collected and analyzed using Zen Blue (Zeiss) or Fiji[28]. We note that FM4-64 staining using the ONIX microfluidics systems is inefficient as the membrane dye appears to bind to the internal plate material. Thus, the majority of images shown in the manuscript were generated using cells immobilized on agarose pads.

**Cross-wall staining and analysis**. For HADA (7-hydroxycoumarin 3-carboxylic acid-amino-D-alanine) labelling[16], spores of the wild type carrying an empty plasmid (SS4), SV55, MB182, SS5, SS414 or MB1120 were loaded into BA04 microfluidic plates (ONIX, CellASIC). Trapped spores were continuously supplied with MYM containing 0.25 or 0.5 mM HADA at 2 psi at 30 °C. Following spore germination, hyphae were allowed to grow by perfusing MYM-HADA at 2 psi for 4–5 h. Prior to image acquisition, MYM-HADA was replaced with MYM and hyphae were visualized using a Zeiss Axio Observer Z1 microscope fitted with a 63x/1.4 Oil BF/DICIII objective. Images were collected using Zen Blue (Zeiss) and analyzed with Fiji[28]. To determine the number of cross-walls per hyphae, the length of individual hyphae originating from single spores was measured manually and the number of cross-walls present in the respective hypha was recorded. Hyphae from at least 20 independent spores were analyzed per replicate ($n = 3$). The distribution of cross-walls per strain and hyphal length was plotted using GraphPad Prism 9.

**Automated western blotting**. Samples of frozen mycelium, originating from 2–5 ml liquid MYM cultures, were resuspended in 0.4 ml ice-cold sonication buffer [20 mM Tris pH 8.0, 5 mM EDTA, 1x EDTA-free protease inhibitors (Sigma Aldrich)] and sonicated ($5 \times 15$ sec on/15 sec off) at 4.5-micron amplitude. Lysates were then centrifuged at $16,000 \times g$ for 15 min at 4 °C to remove cell debris. Total protein concentration was determined using the Bradford assay (Biorad). For anti-FtsZ experiments, 1 μg of total protein from each time point was loaded in triplicate into a microplate (ProteinSimple #043-165) and anti-FtsZ antibody (Polyclonal, Cambridge Research Biochemicals)[12] diluted 1:200. FtsZ levels, originating from the wild type (WT), WT/ftsZ-ypet (SS12), ΔsepX (SV55), ΔsepX/ftsZ-ypet (MB180), ΔdynB (SS2), ΔsepXΔdynB (SV57), ΔsepXΔdynB carrying the empty vector (MB1099) and the complemented ΔsepXΔdynB (MB1103) strains were then assayed using the automated Western blotting machine WES (ProteinSimple, San Jose, CA), according to the manufacturer's guidelines. The ΔftsZ (DU669) strain was used as a negative control. For the detection of mCherry protein fusions, 2.5 μg of total protein and anti-mCherry (Abcam 183628) at a 1:200 dilution was used. mCherry levels, originating from wild-type, ΔftsZ and ΔdynAB strains (as negative controls) or additionally carrying either P_sepX-sepX-mcherry (MB170) or P_ermE*-sepX-mcherry (MB1124, MB1082, MB1092) were then assayed in the same way. For the detection of YPet protein fusions, 1 μg of total protein and anti-GFP (Sigma SAB4301138) at 1:500 dilution was used. For detection of SsgB in WT/ftsZ-ypet (SS12) and ΔsepX/ftsZ-ypet (MB180), 2.5 μg of total protein and anti-SsgB (Polyclonal, Cambridge Research Biochemicals) at 1:100 dilution was used. For detection of WhiA in WT/ftsZ-ypet (SS12) and ΔsepX/ftsZ-ypet (MB180), 0.5 μg of total protein and anti-WhiA[14] (Polyclonal, Cambridge Research Biochemicals) at 1:100 dilution was used. Virtual Western blots were generated using the Compass software for simple western (Version 6.0.0) and uncropped images of all blots are shown in Supplementary Fig. 16.

**SsgB antibody production**. The gene encoding SsgB, vnz_05545, was cloned into pTB146, generating pSS440. Heterologous protein production was induced with 1 mM IPTG at 30 °C for 4 h. Cells were harvested by centrifugation and resuspended in 50 mM Tris-HCl pH 8.0, 1 M NaCl, 5% glycerol, 1% Triton-X. His6-SUMO-SsgB was purified using an HisTrap column (GE Healthcare) and eluted with an increasing imidazole concentration. The protein was dialysed overnight at 4 °C in buffer containing 50 mM Tris-HCl pH 8.0, 150 mM NaCl, 10% glycerol, 1 mM DDT and His6-Upl1 protease at a 1:100 ratio. The released His6-SUMO tag and His6-Upl1 protease were removed by incubation with Ni-NTA affinity agarose beads and a total amount of 2 mg ml$^{-1}$ of untagged SsgB was sent to Cambridge Research Biochemicals (UK) to raise antibodies in rabbits.

**Immunoprecipitation for SepX-mCherry detection**. To be able to detect SepX-mCherry in whole cell lysates, expressed from either the native sepX promoter or the whiE promoter, immunoprecipitation was first conducted with RFP-Trap agarose (Chromotek) prior to automated Western blot analysis. The ΔsepX (SV55), WT/P_ermE*-sepX-mcherry (MB1124), ΔsepX/P_sepX-sepX-mcherry (MB171) and ΔsepX/P_whiE-sepX-mcherry (MB1120) strains were grown in 30 ml MYM liquid cultures at 30 °C and 250 rpm to the required growth stage as confirmed by microscopy. Two cultures for each strain were pooled, pelleted, and washed twice with PBS before resuspension in 1.5 ml lysis buffer (10 mM Tris-Cl pH 7.5, 150 mM NaCl, 0.5 mM EDTA, 1% Triton-X-100, 10 mg/ml lysozyme, 1x EDTA-free protease inhibitors [Sigma Aldrich]). Samples were incubated for 25 min at 37 °C, placed on ice and lysed by sonication using $8 \times 20$ sec at 8 microns with 1 min on ice between each cycle. Lysates were then centrifuged twice at 4 °C and $16,000 \times g$ and the supernatant was retained. Total protein concentration was determined by Bradford assay (Biorad) and the lysates were normalized to ensure equal protein input. 25 μl of binding control agarose beads (Chromotek; bab20), prepared in dilution buffer (10 mM Tris-Cl pH 7.5; 150 mM NaCl, 0.5 mM EDTA) were added to each lysate and left to rotate end-over-end for 1 h at 4 °C. The beads were sedimented at $2,500 \times g$ and 4 °C for 5 min and the supernatant was retained. RFP-Trap agarose beads (Chromotek; rta20) were then added and the lysates were again left to rotate end-over-end for 1 h at 4 °C. The beads were washed with ice-cold lysis buffer according to the manufacturer's instructions and eluted in 50 μl 2x SDS dye. 0.5 μl of each sample was subsequently loaded into a microplate for automated Western blot analysis and mCherry levels were determined as described above.

**Co-immunoprecipitation and mass spectrometry**. Co-immunoprecipitation experiments using SepX as bait ware performed using the SepX-FLAG/DynB-YPet stain (MB942) and the DynB-YPet strain (SS92) as a negative control. Duplicate 30 ml MYM liquid cultures were grown at 30 °C and 250 rpm until the sporulation growth stage, as judged by the detection of DynB-YPet, localized to spore septa using fluorescence microscopy. Cultures were pooled, and formaldehyde (Sigma F8775) was added to a final concentration of 1%. Cultures were further incubated at 30 °C and 250 rpm for 30 mins before quenching the reaction by adding glycine to a final concentration of 125 mM. Cells were pelleted at 4 °C and washed twice with ice-cold PBS, resuspended in 1.5 ml lysis buffer (10 mM Tris-Cl pH 8, 150 mM NaCl, 1% Triton-X-100, 10 mg/ml lysozyme, 1x EDTA-free protease inhibitors [Sigma Aldrich]) and incubated for 25 min at 37 °C. Samples were then

lysed by sonication at 8 × 20 sec at 8 microns with 1 min on ice between each cycle before being centrifuged twice at 4 °C and 16,000 × g, retaining the supernatant. SepX-FLAG was pulled down using the µMACS epitope tag protein isolation kit (FLAG 130-101-591) and eluted using Triethylamine pH 11.8 as described in the manufacturer's instructions. Equal volumes of 2x SDS dye were added to the eluates before heating at 95 °C for 3 min. Equal volumes were then loaded onto a 10% acrylamide gel, ran briefly and each sample was cut out of the gel. Proteins contained within the gel slices were digested with trypsin for 8 h at pH 7.5 and 40 °C and prepared for liquid chromatography-tandem mass spectrometry (LC-MS/MS) as described previously[31] using an Orbitrap Eclipse tribrid mass spectrometer (Thermo Fisher Scientific) and a nanoflow high-performance liquid chromatography (HPLC) system (Dionex Ultimate3000, Thermo Fisher Scientific). MS/MS peaks were analyzed using Mascot v.2.3 (Matrix Science). All Mascot searches were collated and verified with Scaffold (Proteome Software Version 4.0.7). Accepted proteins passed the following threshold in Scaffold: 95% confidence for protein match and a minimum of two unique peptide matches with 95% confidence.

For analysis in R (version 3.6.0)[32], the "Quantitative values" in Scaffold from both the MB942 and SS92 strains, generated from each of three independent experiments were brought together into one dataframe. The function impute.knn (parameters k = 3 and rowmax = 0.9) (package impute) was used to impute the missing values. The values were made into a DGEList object using the function DGEList (package edgeR). Appropriate design and contrast matrices were made using the functions model.matrix and makeContrasts (package limma). Function calcNormFactors (package edgeR) was used to calculate normalisation factors and the voom function (package limma) to log-transform the data and calculate appropriate observation level weights. To fit linear models the ɣmFit function was applied followed by contrasts.fit to compute coefficients (package limma). Differential expression values were calculated using the eBayes function (package limma). The complete data set is available in the Source Data.

**Bacterial adenylate cyclase two-hybrid assays**. To test the interaction between full-length or truncated proteins, coding sequences were amplified using the primers listed in Supplementary Table 1 and cloned in-frame to create the vectors listed in the same table. *E. coli* BTH101 was then co-transformed with 'T25' and 'T18' fusion plasmids. β-galactosidase activity was assayed in biological triplicate in 96-well plates[33].

**Actinobacterial phylogeny and SepX/DynB ortholog identification**. A search set of 673 Actinobacterial genomes were chosen based on annotation as 'reference' or 'representative' at GenBank as of November 2017, along with ten genomes from Phylum Chloroflexi that serve as outgroups. The Chloroflexi genome *Anaerolinea thermophila* UNI-1 was used to root the tree. Amino acid sequences of 37 conserved housekeeping genes were automatically identified, aligned, and concatenated using Phylosift[34]. Model selection was performed using SMS[35] implemented at http://www.atgc-montpellier.fr/phyml/[36], which resulted in the selection of an LG substitution model with gamma distributed rate variation between sites. Phylogenetic reconstruction was performed by RAxML version 8.2.10[37] with 100 rapid bootstraps replicates to assess node support. The tree was visualized and formatted using iTOL[38]. Taxonomic assignments were based on the taxonomy database maintained by NCBI (https://www.ncbi.nlm.nih.gov/Taxonomy/Browser/wwwtax.cgi). SepX and DynB orthologs were identified based on a reciprocal BLAST search (e-value cutoff = 1e-6) of the same set of 673 Actinobacterial genomes, using the *S. venezuelae* SepX/DynB sequences as a query (vnz_14865 and vnz_12105, respectively). For members of the genus *Streptomyces*, this ortholog search was additionally performed on available genomes associated with each species' NCBI Taxonomy ID as of October 2020.

Multiple sequence alignments of SepX homologues from *Streptomyces* and the wider actinobacteria were visualized using Jalview[39] and gene neighbourhood analysis was performed using the webFlaGs server[40].

**Reporting summary**. Further information on research design is available in the Nature Research Reporting Summary linked to this article.

## Data availability statement

Additional data is available on request from the corresponding author. Source data are provided with this paper.

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

## Acknowledgements

We thank Carlo de Oliveira Martins for proteomics analyses. We are also grateful to David Kysela for assistance with phylogenetic analysis, Phil Robinson for taking photographs of *Streptomyces* colonies, and the members of the JIC Bioimaging platform, including Elaine Barclay for help with electron microscopy and Sergio Lopez for support with image analysis. We thank Joseph W. Sallmen for helpful comments on the manuscript and M. Thanbichler for the gift of HADA. This work was funded by the Royal Society (URF\R1\180075) to S.S. and by the BBSRC Institute Strategic Program grant BBS/E/J000PR9791 to the John Innes Centre.

## Author contributions

M.J.B. and S.S. conceived the study, performed experiments and interpreted data, K.A.G. performed phylogenetic analysis, G.C. conducted computational analyses and K.C.F. performed electron microscopy experiments. S.S. supervised the project. M.J.B. and S.S. wrote the manuscript which was approved by all co-authors.

## Competing interests

The authors declare no competing interests.
