## [Peer Review File · Nature Communications]

Hyphal compartmentalization and sporulation in
Streptomyces require the conserved cell division protein SepXREVIEWER COMMENTS

Reviewer #1 (Remarks to the Author):

The manuscript "multicellular growth and sporulation in filamentous actinobacteria require the conserved cell division protein SepX" characterises SepX, a putative membrane protein that seems to co-localise with FtsZ in vegetative and sporulation Z rings, and interacts with DynB, a component of the *Streptomyces* divisome (in this case the sporulation divisome). Discovery a new member of the *Streptomyces* divisome is new and interesting for the researchers working in *Streptomyces* cell division. Authors demonstrate convincingly that SepX participates in sporulation and modify FtsZ ladder dynamics. However, in my opinion, there are some points that can be improved before publication.

MAJOR POINTS

1. The effect of SepX in vegetative hypha septation showed in Fig2a and b and quantified in Fig. 2c is very convincing, due to the differences compared to the wild-type strain, which are complementation when SepX is introduced in trans. However, the image showing the formation of SepX-mCherry rings and ladders colocalising with FtsZ (1 μ m spaced rings) is not so convincing (Fig. 1c, right panel). *Streptomyces* strains usually show autofluorescence at some time points, and is not always easy to separate autofluorescence from the GFP, cherry or YPet signals. For this reason, I miss an image of the *S. venezuelae* wild-type strain without SepX-mCherry take using the same culture and microscopy conditions, to discard autofluorescence artefacts. In addition, why not show a time-lapse of the SepX-mCherry strain? This might be very illustrative.

2. I am not familiar with the YPet protein, and I guess that cross fluorescence mCherry would be discarded by the authors. However, I would appreciate that the authors showed, or indicated as data not shown, that the absence of cross fluorescence between YPet and mCherry was confirmed.

3. In relation with the previous point, line 98: "We were unable to resolve whether SepX localization precedes the arrival of FtsZ, due to the weak fluorescent signal of the SepX-mCherry fusion when produced from its native promoter" If the signal is weak, controls to discard autofluorescence interference are mandatory. Moreover, in Fig. 1c, there are some FtsZ yellow dots that are not present in the SepX-mCherry image (for instance the Z rings at the top of the right hypha). Authors should discuss this, and show more images. Maybe a time-lapse of the SepX-mCherry + FtsZ-YPet can help.

4. Despite that the FtsZ-YPet signal looks very intense, I would suggest to include also control pictures and/or a control time-lapse of the *S. venezuelae* wild-type under the same microscopy conditions to discard autofluorescence artefacts.

5. Title: As all the experimentation was made in *S. venezuelae*, I suggest to change "filamentous actinobacteria" by "*Streptomyces venezuelae*". Authors can speculate in the discussion about the high conservation of SepX and about the possibility that its function in cell division might be general in actinobacteria.

6. Title: "multicellular growth ... require ... SepX". As the SepX mutant still forming multicellular hyphae, this should be changed. I guess that authors refer to colony morphology instead to multicellular growth. The same happens in the last sentence of the abstract (line 34)

7. Differences in spore morphology showed by SEM (Fig. 2d, Fig. 6d-f; supplementary figures 2, 3 and 6) are not clear. These images show very few hyphae. I suggest to include more pictures as supplementary information, and try to quantify these differences. For instance, the variable length of the spores indicated in lines 158-159 might be quantified.

MINOR POINTS

8. Abstract, line 24-27: "... cross-wall formation and the importance of hyphal compartmentalisation for *Streptomyces* development have remained unknown". Not all is unknown, for instance, is well established that FtsZ participates in septa formation in vegetative hyphae. I suggest to say "remains not fully understood" or something like that.

9. The term "cross-wall". When authors use the FM4-64 membrane stain (Fig. 2 for instance, page 6), they talk about cross-walls (line 120 and others), but they are observing membranes, not walls. I understand that the authors assume that if they observe a membrane, it should be associated with a wall, but this is not precise because they are not observing walls. I suggest to talk about "septa", or "membranes", when they refer to the FM4-64 observations.

10. Introduction, lines 47-48: "the physiological significance of cross-wall formation during vegetative growth has remained enigmatic" Why? There are many papers correlating hypha compartmentalisation with morphological and physiological differentiation. There are also many works discussing the importance of septation to protect population against mechanical damage. This sentence needs to be further explained and perhaps referenced.

11. Line 91 "SepX is a 344-amino acid membrane protein" Is the membrane location demonstrated? If not, it should be changed by "putative membrane protein"

12. Line 160. The irregular spore septation show in Supplementary Figure 2g looks very clear. I suggest to move it from supplementary information to Figure 2.

13. Lines 201-203: "these data suggest that SepX-mediated hyphal compartmentalization but not sporulation is crucial for growth and multicellular development in *Streptomyces*" This should be moved to discussion. Moreover, I would change "is crucial for growth and multicellular development" by "is crucial for to determine colony morphology", that is the phenotype showed in Fig. 3f.

14. Lines 213-215: "sporulation in the SepX mutant was much less efficient". Probably this is the case, but is difficult to say with only a few images. More images can be shown in supplementary material to show that this is general. Authors should try to quantify the efficiency of sporulation (see also point 7).

15. Lines 243-245: It was difficult to me to understand why authors say that their results imply that DynB membrane domains are crucial for SepX binding. If I understood well supplementary Figure 5a, it was because they used DynB with and without membrane domains in the hybrid experimentation. This should be clarified in the main text.

16. Line 246: "Importantly, neither SepX nor DynB interacted with RsbN, supporting the notion that SepX specifically associates with DynB (Supplementary Figure 5b)".
I do not see the SepX RsbN interaction in supplementary Fugure 5b.

17. Lines 284-285: Authors used *ermE** to overexpress *sepX*. This might be an approach to increase the low fluorescence signal of *sepX*-mCherry. The *ermE** promoter might be used to overexpress the SepX-mCherry protein (see also point 3).

18. Lines 315-324, phylogenetic analysis of SepX and DynB. Can authors provide an average of SepX identity or similarity? Cell division genes usually are forming operons. Can authors give an idea about conservation of the SepX surrounding chromosomal region? There are some genes that could co-transcript with SepX?

19. Western Blots. I am not familiar with the automated western-blotting. Can authors clarify this methodology? Can they show the raw data (as supplementary information or whatever)? For instance, something that looks strange in Supplementary Figure 1d is that there is a signal in the wt and the *ftsZ* mutant. Is this a non-specific signal? I would appreciate some indication about the reliability of the western blot results shown.

20. Authors should revise reference numeration. For instance, reference 23 (page 347) should be 25, references 24 and 25 should be 23 and 24, reference 28 (page 363) should be reference 27, etc. Page 347 "Santos-Benoit" should be "Santos-Beneit"

21. Lines 347-350. "In agreement with Santos-Benoit and co-workers²³, we did not observe any additional membranous structures in the cytoplasm, also referred to as cross-membranes, which have been proposed to contribute to the subcellular organization of *Streptomyces*^{24,25}"

This sentence should be removed from the discussion. These references can be cited in the introduction when authors describe the complexity of *Streptomyces* cell division. Authors did not make experimentation to detect the cross-membranes described in references 23 and 24. Consequently, does not make sense to discuss that they do not detect cross-membranes. The detection of membranes without peptidoglycan delimitating hypha compartments in young *Streptomyces* hyphae, need the combination of cell wall stains and membrane strains. Otherwise is not possible to define if membranes have or not cell wall/peptidoglycan. Cryo-TEM also can reveal the existence of these membranes, but authors only show TEM pictures from sporulation.

Reviewer #2 (Remarks to the Author):

In this manuscript, Bush et al describe the discovery of SepX an important new member of the bacterial cell division machinery in *Streptomyces venezuelae*, and the first *Streptomyces* protein other than the ubiquitous FtsZ implicated in formation of vegetative cell crosswalls.

In short, the work described here is exciting, technically cutting edge and well performed, and likely to be of interest to anyone interested in bacterial cell biology, especially bacterial cell division, regardless of model organism. I have only minor comments and criticisms. I support acceptance with minor revisions.

- sepX sequence homology. Line 89 onwards: the discovery of sepX as relevant to cell division, and the SepX secondary structure are described. In the discussion later the conservation of the gene in other bacteria is discussed. During this investigation, did the authors also find substantial homology with any other genes of known function? If so does this give any insight into the possible function of the protein?

- The authors state that SepX colocalizes with FtsZ (Line 92) in both sporulation septa and vegetative crosswalls. The supporting evidence is only one image (Figure 1B). Really some sort of colocalization analysis should be performed to properly make this statement.

- I found the effect of sepX deletion on FtsZ localization in vegetative hyphae – extensive appearance of Z-ladders which are normally only present in aerial hyphae - really shocking. Before I read that, if I had to guess based on the preceding results, I would have thought the sepX deletion might lead to no Z-rings, or maybe very misplaced Z-rings, but, wow, this is a very cool result. However, this result is not discussed much. I would really like to hear some of the authors speculations (accepting that it would only be speculation) as to the possible implications of this phenotype for FtsZ regulation and Z-ring positioning in *Streptomyces*. Is it known at all how *Streptomyces* confine their Z-ladders to aerial hyphae? Are there any other gene deletions which lead to a similar unconstrained Z-ladder phenotype?

- Discussion: Overall there is little speculation in the discussion section. I realise this is at least partially a matter of taste, but I feel at least some speculation and discussion of the possible mechanistic role of sepX in cell division would be beneficial. My specific questions are:

- o Is there a known FtsZ membrane anchor for vegetative cross wall formation?

- o Could sepX be a novel membrane anchor for FtsZ?

- o What is the delta sepF phenotype in *S. venezuelae*? Does it bear any similarity (or maybe phenoppose) the delta sepX phenotype

- o If the membrane anchor role is unlikely, do the data point to any other major possible roles for sepX?

Reviewer #3 (Remarks to the Author):

The work in this manuscript nicely describes the characterisation of a new divisome protein in the streptomycetes. Excitingly it is one that is essential for the formation of vegetative crosswalls – a process that is poorly understood in these bacteria – and is also important but not essential for

normal sporulation. The so-called SepX interacts with DynB, presumably in the sporulating hyphae (there are no reports of DynB having a role in vegetative crosswall formation) but neither protein requires the other for proper localisation. Importantly one or both are required for effective FtsZ localisation and septation suggesting that they may act in concert to facilitate sporulation septation.

1. The SepX localisation studies in Fig. 1B seem to suggest much stronger localisation in the vegetative hyphae than in the aerial hyphae. Might this be because there is no DynB-like pathway acting in parallel and thus SepX is THE major crosswall determinant – apart from FtsZ? Whereas in the aerial hyphae SepX seems to act in concert with DynB and so there may be greater flexibility associated with its localisation?

2. Line 111-112- Based on the data presented in Supp Fig 1d, SepX abundance in *ftsZ* mutants seems to be reduced relative to wild type despite being produced from the *ermE** promoter suggesting reduced stability in the absence of FtsZ.

3. The increased vegetative crosswall frequency associated with SepX overexpression is interesting. What happens with sporulation when SepX is overexpressed?

4. The expression of *sepX* from the *whiE* promoter was a clever way of separating protein activity in vegetative vs sporulating cells. Did this restore the normal timing of sporulation or was this still profoundly delayed? (ie – could the timing of sporulation be impacted by the state of the vegetative cells?). While the images presented in Supp Fig 3 suggest more normal sporulation by this strain than the *sepF* mutant, the spore chains in the right hand image still look unusual relative to typical wild type spore chains.

5. Fig 4c/d- Successful septum formation is obvious in the *sepX* mutant but not in the wild type in the images provided. Providing a clearer image here would be helpful for the readers. It is also not clear how 'failed septation' is defined. A large spore where it would have been expected to have at least one additional septation site within it? Or was there some evidence of septation initiation but failure to complete that is not apparent to this reviewer?

6. Lines 225-227- The virtual western blots suggest that FtsZ levels are increased in a *sepX* mutant strain which would make sense given the observation of sporulation septation occurring throughout the mycelium in this mutant. Does loss of *sepX* lead to changes in the levels (expression?) of many proteins typically upregulated during sporulation?

7. Fig 5b- while there is an enrichment of divisome-associated proteins following co-IP with SepX there also appears to be many other proteins. Some commentary on the most represented of these would be appropriate here. For example, what was more abundant than SepF? Given the direct interaction observed with DynB, does the greater abundance of DynA reflect a different stoichiometry of the DynA-DynB levels? Is the expectation that many of the co-IP'd proteins are indirectly associated with SepX, and instead bind e.g. DynB and what has been pulled out is a larger divisome complex?

8. The 2 hybrid experiments suggest that the TM domain of DynB is important for interaction with SepX. An alternative possibility is that loss of the TM domain destabilises DynB so it is no longer available for interaction with SepX. Without an experiment to address this possibility the conclusion that the membrane is critical for SepX binding should be toned down.

9. In a *dynAB* mutant, SepX seems to form much more defined ladders in the sporulating hyphae than in wild type (Fig 6A). Can the authors comment on this?

10. In a *sepX* mutant do the DynB ladders extend through the mycelium as was seen for FtsZ?

11. While there are definite lysed cells present in the images shown in Fig 6D, there also appear to be multiple lysed hyphae in the strains shown in 6E and F as well. Do the colonies in each case look more like the corresponding single mutant strain?

12. Lines 297-300: perhaps consider removing the 'important' designation here – this is a standard complementation experiment.

13. The work would benefit from a short discussion of the genetic context of the *sepX* gene relative to other divisome components in *S. venezuelae* and possibly also other species and genera in which it is found.

Response letter

We thank the reviewers for the critical assessment of our manuscript and the very helpful suggestions on how to improve our work. Please find below our point-by-point response to the requested changes.

REVIEWER COMMENTS

Reviewer #1 (Remarks to the Author):

The manuscript “multicellular growth and sporulation in filamentous actinobacteria require the conserved cell division protein SepX” characterises SepX, a putative membrane protein that seems to co-localise with FtsZ in vegetative and sporulation Z rings, and interacts with DynB, a component of the *Streptomyces* divisome (in this case the sporulation divisome). Discovery a new member of the *Streptomyces* divisome is new and interesting for the researchers working in *Streptomyces* cell division. Authors demonstrate convincingly that SepX participates in sporulation and modify FtsZ ladder dynamics. However, in my opinion, there are some points that can be improved before publication.

MAJOR POINTS

1. The effect of SepX in vegetative hypha septation showed in Fig2a and b and quantified in Fig. 2c is very convincing, due to the differences compared to the wild-type strain, which are complementation when SepX is introduced in trans. However, the image showing the formation of SepX-mCherry rings and ladders colocalising with FtsZ (1µm spaced rings) is no so convincing (Fig. 1c, right panel). *Streptomyces* strains usually show autofluorescence at some time points, and is not always easy to separate autofluorescence from the GFP, cherry or YPet signals. For this reason, I miss an image of the *S. venezuelae* wild-type strain without SepX-mCherry take using the same culture and microscopy conditions, to discard autofluorescence artefacts. In addition, why not show a time-lapse of the SepX-mCherry strain? This might be very illustrative.

We agree with the reviewer that *Streptomyces* display some autofluorescence, in particular in the green/yellow wavelength range, which is often visible as individual foci within cell filaments. However, we generally do not observe such autofluorescence artefacts when using mCherry as a fluorescent reporter. The still images shown in Figure 1c were originally acquired as part of a time-lapse experiment. We now provide the complete time-series in Supplementary Movie 1, which also shows a larger field of view with several hyphae undergoing sporulation septation. Furthermore, we included additional images of control experiments with single-labelled strains in Supplementary Figure 1, demonstrating the specificity of the fluorescence signals for SepX-mCherry and FtsZ-YPet. The manuscript text has been modified accordingly: Line 97-102). Please see also response to Reviewer 2, comment #2.

2. I am not familiar with the YPet protein, and I guess that cross fluorescence mCherry would be discarded by the authors. However, I would appreciate that the authors showed, or indicated as data not shown, that the absence of cross fluorescence between YPet and mCherry was confirmed.

YPet is a yellow-fluorescent GFP derivative. The reviewer is correct, we have confirmed that there is no crosstalk of fluorescence when YPet and mCherry-labelled proteins are sequentially excited, using excitation and emission bandwidths to minimise spectral bleed-through. Furthermore, our microscope is equipped with single band-pass filters to minimize potential cross talk between YPet and mCherry fluorescence. We have added control images from single-labelled strains showing the absence of bleed-through fluorescence when imaging FtsZ-YPet and SepX-mCherry (Supplementary Figure 1).

3. In relation with the previous point, line 98: “We were unable to resolve whether SepX localization precedes the arrival of FtsZ, due to the weak fluorescent signal of the SepX-mCherry fusion when produced from its native promoter” If the signal is weak, controls to discard autofluorescence interference are mandatory. Moreover, in Fig. 1c, there are some FtsZ yellow dots that are not present in the SepX-mCherry image (for instance the Z rings at the top of the right hypha). Authors should discuss this, and show more images. Maybe a time-lapse of the SepX-mCherry + FtsZ-YPet can help.

Please see response to comment 1, 2 and 4 and response to comment #2 by Reviewer 2. We believe that the new additional data, including a time-lapse movie, control images and a co-localization analysis now fully support our conclusion that SepX and FtsZ co-localise at cross-walls and sporulation septa and demonstrate that the fluorescent signals detected for each protein fusion are specific. It was not clear to us to which “yellow FtsZ dots” the reviewer referred to in Figure 1c. It is possible that those dots were the result of a transient accumulation of FtsZ-YPet and we hope that this matter has been resolved by providing the corresponding time-lapse movie (Supplementary Movie 1).

4. Despite that the FtsZ-YPet signal looks very intense, I would suggest to include also control pictures and/or a control time-lapse of the *S. venezuelae* wild-type under the same microscopy conditions to discard autofluorescence artefacts.

We agree with the reviewer that the FtsZ-YPet signal is readily detectable. We have added control images obtained from a strain that does not produce the FtsZ-YPet fusion to demonstrate the absence of autofluorescence under the conditions used in this study (Supplementary Figure 1). We note that the FtsZ-YPet fusion is a well-established and widely-used construct to visualize and investigate FtsZ localization in *Streptomyces* (*Nat Commun* (2021) 12:5222; *eLife* (2021) 10; *Mol Micro* (2019) 112:1; *PNAS* (2017) 114:30; *J. Vis. Exp.* (108), e53863).

5. Title: As all the experimentation was made in *S. venezuelae*, I suggest to change “filamentous actinobacteria” by “*Streptomyces venezuelae*”. Authors can speculate in the discussion about the high conservation of SepX and about the possibility that its function in cell division might be general in actinobacteria.

We have changed the title to “Hyphal compartmentalization and sporulation in *Streptomyces* require the conserved cell division protein SepX” (please see also response to comment #6 below). We would prefer to keep “*Streptomyces*” rather than *Streptomyces venezuelae* in the title (and the abstract, Line 35) as it is clear from our phylogenetic analysis that SepX is conserved within the genus. Furthermore, we found that expressing the *sepX* homologue from *Streptomyces coelicolor* in *S. venezuelae* can complement the Δ *sepX* mutant phenotype and restores normal growth and sporulation (Response Figure 1). This clearly indicates that SepX function is conserved among *Streptomyces* species.

Response Figure 1: Cross-complementation experiment. Expression of *sepX* from *S. coelicolor* *in trans* in the *S. venezuelae* $\Delta sepX$ mutant can restore vegetative septa formation and wildtype-like sporulation. Vegetative septa were visualized using the fluorescent membrane dye FM4-64 (scale bars 10 μm). Spore chains were images by cryo-SEM (scale bars: 2 μm).

6. Title: “multicellular growth ... require ... SepX”. As the SepX mutant still forming multicellular hyphae, this should be changed. I guess that authors refer to colony morphology instead to multicellular growth. The same happens in the last sentence of the abstract (line 34)

We have replaced “multicellular growth” with “hyphal compartmentalization” in the title and changed “multicellular” to “cellular in the abstract (line 35) to clarify the importance of SepX in cell division and morphogenesis in *Streptomyces*.

7. Differences in spore morphology showed by SEM (Fig. 2d, Fig. 6d-f; supplementary figures 2, 3 and 6) are not clear. These images show very few hyphae. I suggest to include more pictures as supplementary information, and try to quantify these differences. For instance, the variable length of the spores indicated in lines 158-159 might be quantified.

We now provide uncropped SEM images for all the SEM images shown in this work, including additional SEM images for each of the presented strains in Supplementary Figure 9-12.

As suggested, we have quantified the described differences in spore morphology for the wildtype, the $\Delta sepX$ mutant and the complemented $\Delta sepX$ mutant using automated cell sizing and counting. Specifically, we determined the diameter of 100,000 spores per strain (in biological triplicate). We found that spores generated by the $\Delta sepX$ mutant consistently displayed a larger and more variable spore volume compared to the wildtype and the complemented mutant. This finding is indicative of defective sporulation-septation, which is consistent with the aberrant spore sizes observed by electron microscopy. We have added these new results of this analysis to Figure 2 and modified the text accordingly (Line 163-165).

MINOR POINTS

8. Abstract, line 24-27: “... cross-wall formation and the importance of hyphal compartmentalisation for *Streptomyces* development have remained unknown”. Not all is unknown, for instance, is well

established that FtsZ participates in septa formation in vegetative hyphae. I suggest to say “remains not fully understood” or something like that.

As suggested, we have revised this sentence, which reads now “... specific determinants for cross-wall formation and the importance of hyphal compartmentalization for *Streptomyces* development have remained largely unknown.” (Line 28).

9. The term “cross-wall”. When authors use the FM4-64 membrane stain (Fig. 2 for instance, page 6), they talk about cross-walls (line 120 and others), but they are observing membranes, not walls. I understand that the authors assume that if they observe a membrane, it should be associated with a wall, but this is not precise because they are not observing walls. I suggest to talk about “septa”, or “membranes”, when they refer to the FM4-64 observations.

Good point. We have now clarified this in the text and state that FM4-64 staining was used to visualize vegetative septa (Line 124), while HADA staining was used to detect cross-walls. We note that we found that both fluorescent labelling methods, using either FM4-64 or HADA, produced a similar staining pattern.

10. Introduction, lines 47-48: “the physiological significance of cross-wall formation during vegetative growth has remained enigmatic” Why? There are many papers correlating hypha compartmentalisation with morphological and physiological differentiation. There are also many works discussing the importance of septation to protect population against mechanical damage. This sentence needs to be further explained and perhaps referenced.

We agree with the reviewer that the presence of cross-walls has been shown to provide protection against mechanical stress. We have toned down the statement and referenced relevant work. However, to the best of our knowledge, there are currently no other publications providing direct experimental evidence that cross-walls are required for normal development in *Streptomyces*. The physiological significance of cross-walls has previously been impossible to dissect because the only mutant known to completely lack cross-walls is the *ftsZ* null mutant, which also lacks sporulation septa. Thus, the specific contribution of cross-walls for growth and development had remained unknown. With the identification of SepX we have been able to firmly demonstrate that cross-wall mediated hyphal compartmentalization and not sporulation-associated septation is crucial for fitness and cellular development of *Streptomyces*.

11. Line 91 “SepX is a 344-amino acid membrane protein” Is the membrane location demonstrated? If not, it should be changed by “putative membrane protein”

We have inserted “putative” as suggested (Line 91).

12. Line 160. The irregular spore septation show in Supplementary Figure 2g looks very clear. I suggest to move it from supplementary information to Figure 2.

As suggested, we have moved the TEM images from the Supplementary Figure 2 to the main Figure 2. The cryo-SEM images previously shown in Figure 2d, are now part of the revised Supplementary Figure 2. The text in the manuscript has been revised accordingly (Line 158-165).

13. Lines 201-203: “these data suggest that SepX-mediated hyphal compartmentalization but not sporulation is crucial for growth and multicellular development in *Streptomyces*” This should be

moved to discussion. Moreover, I would change “is crucial for growth and multicellular development” by “is crucial for to determine colony morphology”, that is the phenotype showed in Fig. 3f.

We have modified this sentence to read “Collectively, these data suggest that SepX-mediated hyphal compartmentalization but not sporulation is crucial for growth and cellular development in *Streptomyces*” (Line204-206). We appreciate the reviewer’s opinion but feel that this sentence is also appropriate in its current position as it summarises the findings presented in this section. The lack of cross-walls directly affects cellular development in *Streptomyces*, which results in altered growth and colony morphology.

14. Lines 213-215: “sporulation in the SepX mutant was much less efficient”. Probably this is the case, but is difficult to say with only a few images. More images can be shown in supplementary material to show that this is general. Authors should try to quantify the of sporulation (see also point 7).

We thank the reviewer for this suggestion. We have performed additional experiments to clarify the statement about the reduced sporulation efficiency of the $\Delta sepX$ mutant (see also response to comment #7). We used automated cell counting to quantify the total number of spores produced by the wildtype, the $\Delta sepX$ mutant and the complemented mutant in a given volume. Three independent spore stocks of each strain grown on solid medium were prepared and subsequently analysed with a Multisizer 4e Coulter Counter. Our results, now mentioned on Line 218-220, show that the $\Delta sepX$ mutant produces 50% less spores compared to the wildtype after three days of incubation, a time-period that is sufficient for wild-type *S. venezuelae* to complete sporulation (Bush et al. (2019) mBio 10:e02812-18) (Supplementary Figure 4c).

15. Lines 243-245: It was difficult to me to understand why authors say that their results imply that DynB membrane domains are crucial for SepX binding. If I understood well supplementary Figure 5a, it was because they used DynB with and without membrane domains in the hybrid experimentation. This should be clarified in the main text.

The reviewer is correct. We have used a DynB variant that lacked the transmembrane segments (DynB Δ TM) in our two-hybrid experiments. DynB Δ TM failed to bind SepX. We have clarified this in the text (Line 254-257). When we revised the manuscript, we also noted that the original Supplementary Figure 5a missed the relevant data. We have fixed our error and now provide all relevant data in Supplementary Figure 5a.

16. Line 246: “Importantly, neither SepX nor DynB interacted with RsbN, supporting the notion that SepX specifically associates with DynB (Supplementary Figure 5b)”. I do not see the SepX RsbN interaction in supplementary Fugure 5b.

The reviewer might have missed this. The results of the SepX-RsbN interaction studies are shown in the Supplementary Figure 5b. It is the second bar from the left.

17. Lines 284-285: Authors used ermE* to overexpress sepX. This might be an approach to increase the low fluorescence signal of sepX-mCherry. The ermE* promoter might be used to overexpress the SepX-mCherry protein (see also point 3).

Please see responses to previous comments above regarding Figure 1. We believe we have been able to clarify the localization pattern of SepX-mCherry by providing a time-lapse movie, appropriate

control images and by analysing the fluorescence intensities of SepX-mCherry and FtsZ-YPet during sporulation septation (revised Figure 1, Supplementary Figure 1, Supplementary Movie 1).

18. Lines 315-324, phylogenetic analysis of SepX and DynB. Can authors provide an average of SepX identity or similarity? Cell division genes usually are forming operons. Can authors give an idea about conservation of the SepX surrounding chromosomal region? There are some genes that could co-transcript with SepX?

The amino acid sequence length of SepX homologs from *Streptomyces* and the wider actinobacteria varies greatly (300 to 1000 amino acids). Due to the large gaps present within the multiple sequence alignment, providing an average sequence identity or similarity will be less informative.

To obtain a better understanding about the level of sequence conservation across different SepX homologs, we have analyzed multiple sequence alignments of representative SepX homologs from *Streptomyces* (n=58) and the wider actinobacteria (n=251) and used Jalview (Waterhouse et al. *Bioinformatics*, 25(9), 2009, 1189-1191) to identify regions of high sequence identity. This analysis revealed that the predicted transmembrane regions are conserved across all SepX homologs, supporting the idea that the role of SepX in cell division may require the association with the cytoplasmic membrane. Our analysis further revealed, that there are several residues in the C-terminal cytoplasmic domain that display high sequence conservation in SepX homologs from *Streptomyces*. The importance of these residues for SepX function is not known. We have included this additional analysis as Supplementary Figure 8 and briefly discuss these new results in text (Line 336-340).

We have further inspected the conservation of the *sepX* gene neighbourhood in genomes of *streptomycetes* and representative examples of the wider *actinomycetes*, using the webFlaGs server (Saha et al. *Bioinformatics*, 37(9), 2021, 1312–1314). The gene encoding SepX (*vnz_14865*) does not lie in an operon or in the gene neighbourhood of genes known to encode cell division proteins. The output of this analysis is shown in Supplementary Figure 8c and we briefly discuss the results in the revised manuscript (Line 408-410).

19. Western Blots. I am not familiar with the automated western-blotting. Can authors clarify this methodology? Can they show the raw data (as supplementary information or whatever)?

The automated Western blot platform is a commercial system that is based on capillary electrophoresis (<https://www.proteinsimple.com/wes.html>). Western blot assays are performed in capillaries where SDS/heat-denatured proteins are separated by apparent molecular weight, immobilized covalently via a UV sensitive coating in the capillary, detected directly in the capillary by primary and secondary antibodies and analyzed by an integrated software. The data can be visualized as chemiluminescence spectra or shown as “virtual Western blots”.. This technique provides excellent reproducibility compared to traditional Western blots and it suitable for quantitative analysis. We also note that the WES is a well-established methodology that has been used widely in the field, e.g. Feeney et al. *mBio*. 2017 Jun 13;8(3):e00815-17; Bush et al. *Mol Microbiol*. 2017 Jun;104(5):700-711; Bush et al. *mBio* 2019 Feb 5;10(1):e02812-18 and Gallagher et al. *Mol Cell* 2020 Feb 6;77(3):586-599.e6.

We now provide uncropped images of all virtual Western blots shown in the manuscript in Supplementary Figure 13. To view the raw data and the chemiluminescence spectra, the reader will have to install an additional programme (<https://www.proteinsimple.com/compass/downloads>) and thus, we believe that showing the virtual Western blots is more practical. We are happy to include the

raw data as a source file if the Editor considers this to be crucial but would like to suggest sharing the raw data files upon request.

For instance, something that looks strange in Supplementary Figure 1d is that there is a signal in the wt and the *ftsZ* mutant. Is this a non-specific signal? I would appreciate some indication about the reliability of the western blot results shown.

The reviewer is correct, this signal in Supplementary Figure 1d is a result of unspecific binding of the mCherry antibody because this signal is also present in cell lysates derived from the untagged wild-type and the Δ *ftsZ* control. We have now clarified this in the figure and the figure legend. As stated in the figure legends, virtual Western blots are the result of biological duplicate experiments.

20. Authors should revise reference numeration. For instance, reference 23 (page 347) should be 25, references 24 and 25 should be 23 and 24, reference 28 (page 363) should be reference 27, etc. Page 347 “Santos-Benoit” should be “Santos-Beneit”

Thank you for spotting this. The reference list has been revised.

21. Lines 347-350. “In agreement with Santos-Benoit and co-workers²³, we did not observe any additional membranous structures in the cytoplasm, also referred to as cross-membranes, which have been proposed to contribute to the subcellular organization of *Streptomyces*^{24,25}”. This sentence should be removed from the discussion. These references can be cited in the introduction when authors describe the complexity of *Streptomyces* cell division. Authors did not make experimentation to detect the cross-membranes described in references 23 and 24. Consequently, does not make sense to discuss that they do not detect cross-membranes. The detection of membranes without peptidoglycan delimitating hypha compartments in young *Streptomyces* hyphae, need the combination of cell wall stains and membrane strains. Otherwise is not possible to define if membranes have or not cell wall/peptidoglycan. Cryo-TEM also can reveal the existence of these membranes, but authors only show TEM pictures from sporulation.

We have deleted this sentence.

Reviewer #2 (Remarks to the Author):

In this manuscript, Bush et al describe the discovery of SepX an important new member of the bacterial cell division machinery in *Streptomyces venezuelae*, and the first *Streptomyces* protein other than the ubiquitous FtsZ implicated in formation of vegetative cell crosswalls. In short, the work described here is exciting, technically cutting edge and well performed, and likely to be of interest to anyone interested in bacterial cell biology, especially bacterial cell division, regardless of model organism. I have only minor comments and criticisms. I support acceptance with minor revisions.

Thank you!

1. sepX sequence homology. Line 89 onwards: the discovery of sepX as relevant to cell division, and the SepX secondary structure are described. In the discussion later the conservation of the gene in other bacteria is discussed. During this investigation, did the authors also find substantial homology with any other genes of known function? If so does this give any insight into the possible function of the protein?

Our bioinformatic analyses (including the use of AlphaFold2, the Dali server, Phyre2 and I-Tasser) did not reveal any sequence or structural homology of SepX to other proteins with known function. We have added this information to the revised manuscript (Line 93-94).

2. The authors state that SepX colocalizes with FtsZ (Line 92) in both sporulation septa and vegetative crosswalls. The supporting evidence is only one image (Figure 1B). Really some sort of colocalization analysis should be performed to properly make this statement.

The still images shown in Figure 1 were taken from a time-lapse movie. We now provide the full time-lapse series as Supplementary Movie 1. This movie also provides a larger field of view with several hyphae that show co-localization of FtsZ-YPet and SepX-mCherry at vegetative and sporulation septa. Furthermore, we have determined the fluorescence intensity profiles for FtsZ-YPet and SepX-mCherry in sporulating hyphae, showing that fluorescence peak intensities for both protein fusions spatially and temporarily overlap (Figure 1e and Supplementary Figure 1b).

3. I found the effect of *sepX* deletion on FtsZ localization in vegetative hyphae – extensive appearance of Z-ladders which are normally only present in aerial hyphae - really shocking. Before I read that, if I had to guess based on the preceding results, I would have thought the *sepX* deletion might lead to no Z-rings, or maybe very misplaced Z-rings, but, wow, this is a very cool result. However, this result is not discussed much. I would really like to hear some of the authors speculations (accepting that it would only be speculation) as to the possible implications of this phenotype for FtsZ regulation and Z-ring positioning in *Streptomyces*. Is it known at all how *Streptomyces* confine their Z-ladders to aerial hyphae? Are there any other gene deletions which lead to a similar unconstrained Z-ladder phenotype?

We agree, the sudden burst of Z-ladders in the *sepX* mutant is visually quite striking. How *Streptomyces* confine Z-ladders to the hyphal tip compartment is not understood. It has been suggested that the assembly of Z-ladders is preceded by the formation of a “basal septum” at the base of the sporogenic tip compartment. However, experimental evidence for the presence of such a septum is still missing. We are not aware of any other gene deletion that results in a similar spreading of Z-ladders through the mycelium.

We speculate that the absence of hyphal compartmentalization affects the expression of key developmental regulators and/or the compartment-specific accumulation of proteins or signals required for the coordinated progression through the developmental life cycle. For example, we found that protein levels for FtsZ, the FtsZ-interacting protein SsgB and the transcriptional regulator WhiA, which co-controls the entry into sporulation, are increased in the *sepX* mutant compared to the wildtype (Supplementary Figure 4f). It should be noted that we find that overexpression of just *ftsZ* from a constitutive promoter does not lead to the abundant appearance of Z-ladders (unpublished data). Although our WES results imply that key components for division assembly are more abundant in *sepX*-deficient hyphae, other factors required for the formation of division-competent Z-rings may not be produced at equivalent levels in the *sepX* mutant. We don't yet fully understand how Z-rings are positioned in *Streptomyces*, but this is something we would like to uncover in the future.

We have extended the discussion and included some speculation on the possible implications when hyphae grow without cross-walls (Line 373-386, please see also response to comment #4 below).

4. Discussion: Overall there is little speculation in the discussion section. I realise this is at least partially a matter of taste, but I feel at least some speculation and discussion of the possible mechanistic role of sepX in cell division would be beneficial.

As suggested, we have added some speculations about the possible role of SepX in cell division to the discussion (Line 408-415).

My specific questions are:

o Is there a known FtsZ membrane anchor for vegetative cross wall formation?

No, there is no known FtsZ membrane anchor that for vegetative cross-wall formation. A potential candidate for this role could be SepF, which has been shown to function as an alternative membrane anchor for FtsZ in *B. subtilis* (Yu et al. MBio (2021) Volume 12 Issue 1 e02964-20). We previously showed that SepF co-localizes with FtsZ at vegetative cross-walls but also at sporulation septa in *Streptomyces* (Schlimpert et al, PNAS (2017) 114:30). However, the exact function of SepF has not yet been determined in *Streptomyces*.

o Could sepX be a novel membrane anchor for FtsZ?

That is possible. However, we have not been able to demonstrate a direct interaction between SepX and FtsZ to support this idea. It is also conceivable that SepX mediates the recruitment of FtsZ to the membrane via a yet unidentified interaction partner. Interestingly, analysis of SepX homologs from *Streptomyces* and the wider actinobacteria revealed that residues predicted to be involved in membrane binding, are conserved across different bacterial species (Supplementary Figure 8a-b). Thus, we believe that the interaction of SepX with the cytoplasmic membrane is important for its function in cell division.

o What is the delta sepF phenotype in *S. venezuelae*? Does it bear any similarity (or maybe phenoppose) the delta sepX phenotype

There is no $\Delta sepF$ deletion mutant described for *Streptomyces* so far. Notably, work in related Actinobacteria, including *Mycobacterium* (Gola et al., *Mol. Microbiol.* 97:560-576 (2015); Gupta et al. *Micobiology* 161:1627-1638 (2015) and *Corynebacterium* (Sogues et al. *Nat. Commun.* (2020) 11:1641) showed that FtsZ and SepF are interdependent and both essential for Z-ring and septum formation. It is therefore conceivable that SepF fulfils a similar essential function in *Streptomyces* cell division and deletion of *sepF* is likely to result in a phenotype that resembles the *ftsZ* null mutant, which is unable to form cross-walls and sporulation septa.

o If the membrane anchor role is unlikely, do the data point to any other major possible roles for sepX?

No, currently not.

Reviewer #3 (Remarks to the Author):

The work in this manuscript nicely describes the characterisation of a new divisome protein in the streptomycetes. Excitingly it is one that is essential for the formation of vegetative crosswalls – a process that is poorly understood in these bacteria – and is also important but not essential for normal sporulation. The so-called SepX interacts with DynB, presumably in the sporulating hyphae (there are no reports of DynB having a role in vegetative crosswall formation) but neither protein requires the

other for proper localization. Importantly one or both are required for effective FtsZ localization and septation suggesting that they may act in concert to facilitate sporulation septation.

1. The SepX localization studies in Fig. 1B seem to suggest much stronger localization in the vegetative hyphae than in the aerial hyphae. Might this be because there is no DynB-like pathway acting in parallel and thus SepX is THE major crosswall determinant – apart from FtsZ? Whereas in the aerial hyphae SepX seems to act in concert with DynB and so there may be greater flexibility associated with its localization?

We agree with the reviewer, SepX-mCherry localization is much better discernible at vegetative septa. Based on our results, we do believe that SepX is the major determinant for cross-wall formation. During vegetative growth *dynB* is not expressed (Schlimpert et al, *PNAS* (2017) 114:30), therefore the role of SepX is independent of DynB at cross-walls. The dynamics of cell division leading to cross-wall formation are not understood and it is conceivable, as the reviewer suggested that in sporulating hyphae SepX localization is more flexible due to the presence of DynB and possibly other sporulation-specific cell division factors. We now comment on the localization pattern of SepX in the discussion (Line 359-362).

2. Line 111-112- Based on the data presented in Supp Fig 1d, SepX abundance in *ftsZ* mutants seems to be reduced relative to wild type despite being produced from the *ermE** promoter suggesting reduced stability in the absence of FtsZ.

We agree with the reviewer that SepX abundance seems to be lower in the Δ *ftsZ* mutant. Although we cannot exclude the possibility that SepX is turned-over more rapidly in *ftsZ*-deficient hyphae, judging by the detected signal corresponding to free mCherry, it appears that *sepX* expression rather than degradation might be reduced in the Δ *ftsZ* mutant. The reason for this is unknown.

3. The increased vegetative crosswall frequency associated with SepX overexpression is interesting. What happens with sporulation when SepX is overexpressed?

There is no apparent effect on sporulation when *sepX* is overexpressed and the spore chains formed under these conditions have a wildtype-like morphology (Line 143-146, Supplementary Figure 2b).

4. The expression of *sepX* from the *whiE* promoter was a clever way of separating protein activity in vegetative vs sporulating cells. Did this restore the normal timing of sporulation or was this still profoundly delayed? (ie – could the timing of sporulation be impacted by the state of the vegetative cells?). While the images presented in Supp Fig 3 suggest more normal sporulation by this strain than the *sepF* mutant, the spore chains in the right hand image still look unusual relative to typical wild type spore chains.

The macroscopic development of colonies formed by the conditional Δ *sepX* mutant is similar to the *sepX* deletion mutant, and thus suggests that the timing of sporulation is affected when *sepX* expression is driven from the *whiE* promoter. In this background, hyphae are largely devoid of cross-walls (Supplementary Figure 3b-d). We found that in the absence of cross-walls, the abundance of several proteins required for sporulation-specific cell division was increased compared to the wildtype (Supplementary Figure 4f). Our results therefore support the idea that the state of the vegetative

hyphae influences the progression through the developmental programme, including the timing of sporulation.

To support our statement that spores produced in the *sepX* mutant expressing *sepX-mcherry* from the *whiE* and the *sepX* promoter are similar in size to the wildtype, we have determined the size of spores produced by the complemented $\Delta sepX$ mutant (*sepX_p-sepX-mcherry*) and the conditional $\Delta sepX$ mutant (*whiE_p-sepX-mcherry*). Measuring 100,000 spores per strain (n=3), we found that expression of *sepX* from either the *whiE* or its native promoter largely restores wildtype-like spore morphology (Line 201-203 and Supplementary Figure 3f). Furthermore, we provide additional SEMs for each strain in Supplementary Figure 11.

5. Fig 4c/d- Successful septum formation is obvious in the *sepX* mutant but not in the wild type in the images provided. Providing a clearer image here would be helpful for the readers. It is also not clear how 'failed septation' is defined. A large spore where it would have been expected to have at least one additional septation site within it? Or was there some evidence of septation initiation but failure to complete that is not apparent to this reviewer?

We have replaced Figure 4c with a clearer set of representative images to show regular and complete sporulation septation in the wildtype and clarified in the text how 'failed septation' in the $\Delta sepX$ mutant was defined. It is difficult to visualize incomplete constrictions by light microscopy. However, the sporulation septation phenotype shown in Figure 4c/d is in line with electron micrographs of spore chains produced by the wildtype and the $\Delta sepX$ mutant, which we show in Figure 2f and Supplementary Figure 2h-l. These images revealed that spore chains in the $\Delta sepX$ mutant often displayed asymmetric and incomplete constrictions, suggesting that the septation process had been initiated but was prematurely aborted (Line 163-169 and Line 218-223).

6. Lines 225-227- The virtual western blots suggest that FtsZ levels are increased in a *sepX* mutant strain which would make sense given the observation of sporulation septation occurring throughout the mycelium in this mutant. Does loss of *sepX* lead to changes in the levels (expression?) of many proteins typically upregulated during sporulation?

Good question. We have performed additional WES experiments and found that in addition to FtsZ, the abundance of the sporulation-specific cell division protein SsgB (required for Z-ring formation) and the transcriptional regulator WhiA (essential for sporulation) was increased in the $\Delta sepX$ mutant compared to the wildtype. This suggests that the absence of *sepX*, and consequently the lack of cross-walls, affect the abundance of both structural and regulatory proteins involved in cell division, which in turn could influence the spatial and temporal regulation of sporulation. We have included the new WES data in Supplementary Figure 4 and discuss the results in the revised manuscript (Line 232-236 and Line 373-382).

7. Fig 5b- while there is an enrichment of divisome-associated proteins following co-IP with SepX there also appears to be many other proteins. Some commentary on the most represented of these would be appropriate here. For example, what was more abundant than SepF? Given the direct interaction observed with DynB, does the greater abundance of DynA reflect a different stoichiometry of the DynA-DynB levels? Is the expectation that many of the co-IP'd proteins are indirectly associated with SepX, and instead bind e.g. DynB and what has been pulled out is a larger divisome complex?

We agree with the reviewer that the co-IP results likely present the composition of the larger divisome complex rather than direct protein interactions with SepX and that SepX becomes part of this network through the association with DynB. We also note that the experiments were performed using a cross-linking agent which could have promoted the purification of larger protein complexes. The stoichiometry of DynA and DynB *in vivo* is not known. Furthermore, the mass-spec analysis performed was not quantitative and thus, the results don't allow us to deduce the stoichiometry of DynA and DynB based on the detected peptides.

Among the proteins that were most enriched apart from SepX were proteins predicted to be involved in metabolism and transport. It is conceivable that these proteins are highly abundant in the cell and reside in close proximity to SepX, which resulted in the observed enrichment of these proteins in our analysis. To clarify this point, we have provided more information on the co-IP analysis in the text (Line 270-273 and Line 278-279) and included a list of all proteins identified in the co-IP analysis in the Source Data.

8. The 2 hybrid experiments suggest that the TM domain of DynB is important for interaction with SepX. An alternative possibility is that loss of the TM domain destabilises DynB so it is no longer available for interaction with SepX. Without an experiment to address this possibility the conclusion that the membrane is critical for SepX binding should be toned down.

We agree with the reviewer's comment and have toned down our conclusion (Line 254-257).

9. In a *dynAB* mutant, SepX seems to form much more defined ladders in the sporulating hyphae than in wild type (Fig 6A). Can the authors comment on this?

Good point. The reason for the more defined localization of SepX-mCherry is that production of the protein fusion was driven from a constitutively promoter in this strain. To provide more uniform images, we have replaced this image with an image of the Δ *dynAB* mutant expressing *sepX-mcherry* from its native promoter (MB1270). Under these conditions, SepX-mCherry accumulation at division septa in sporulating hyphae is similar to the wildtype.

10. In a *sepX* mutant do the DynB ladders extend through the mycelium as was seen for FtsZ?

This is a great question. We have performed additional time-lapse experiments to visualize the subcellular localization of DynB-YPet in the Δ *sepX* mutant. In general, fluorescence emitted from the DynB-YPet fusion is quite dim, requiring longer exposure times which turned out to cause phototoxicity. Based on the limited localization data we were able to collect, we found that DynB-YPet ladders do not spread through the mycelium like FtsZ-YPet ladders. It should be noted that *dynB* expression is controlled by a sporulation-specific promoter whereas *ftsZ* expression is driven by at least five vegetative and sporulation-specific promoters, leading to higher FtsZ levels in the *sepX* mutant (see also response to comment #6).

11. While there are definite lysed cells present in the images shown in Fig 6D, there also appear to be multiple lysed hyphae in the strains shown in 6E and F as well. Do the colonies in each case look more like the corresponding single mutant strain?

The reviewer is correct, we do see some cell lysis in the complemented mutants, but this is much more pronounced in the Δ *sepX* Δ *dynB* double mutant. Complementing the Δ *sepX* Δ *dynB* double mutant with

either *sepX* or *dynB* results in a colony phenotype that largely restores the appearance of the corresponding single mutant in size and shape (Response Figure 2). We note, however, that although green spore pigmentation is not restored, spore chains produced by the complemented mutants reconstitute the sporulation defect in the corresponding single mutants (Figure 6D).

Response Figure 2: Colony morphology of the $\Delta sepX\Delta dynB$ (MB1099) double mutant and the corresponding single mutants (SS2, MB1101, MB1103, SV55). Strains were grown for 3 days on MYM agar. Scale bars: 1 mm.

To give a better representation of the described phenotypes, we now provide the original (uncropped) images for each of the SEM images shown in the manuscript, and additional SEM images in Supplementary Figure 11.

12. Lines 297-300: perhaps consider removing the ‘important’ designation here – this is a standard complementation experiment.

Agreed. We have removed the word “importantly” from the sentence (Line 311).

13. The work would benefit from a short discussion of the genetic context of the *sepX* gene relative to other divisome components in *S. venezuelae* and possibly also other species and genera in which it is found.

The manuscript now contains a brief statement about the genetic context of *sepX* in the genome of *S. venezuelae* and the wider actinobacteria (Line 408-410) and we added Supplementary Figure 8c to illustrate our findings (please see also response to Reviewer 1/comment #18). Briefly, we used the webFlaGs server (Saha CK et al. Bioinformatics, 37(9), 2021, 1312–1314) to analyse the conservation of the *sepX* gene neighbourhood. *sepX* is an orphan gene that is not encoded in the vicinity of the *dcw* cluster or other genes known to encode cell division protein in *Streptomyces* and the wider actinobacteria.

REVIEWERS' COMMENTS

Reviewer #1 (Remarks to the Author):

The revised version of the manuscript has addressed most of my comments. The work provides new insights about *Streptomyces* cell division and deserves publication. I have two final comments to be considered prior publication:

1. I guess that the next sentence (line 98) needs to be rephrased (see my comment 1, and reviewer's 2 comment 2): "Microscopic analysis of this dual-labelled strain by time-lapse microscopy revealed that SepX-mCherry colocalizes with FtsZ at both vegetative cross-walls and sporulation septa (Figure 1c, d, Supplementary Figure 1b and Supplementary Movie 1)".

In Figure 1b and the new Supplementary Movie 1, there are several Z-rings that do not colocalise with SepX (see examples marked with arrows in a time-lapse frame; the PDF with the image was sent to the Editor). The fact that not all Z-rings co-localise with SepX should be indicated and discussed in the manuscript.

This does not detract the results. As the authors indicate, they could not "resolve whether SepX localization precedes the arrival of FtsZ". This, together with the fact that not all Z-rings colocalise with SepX, indicates that further work will be necessary to fully understand the interaction between these two proteins. However, authors demonstrated that they colocalise, at least at some point, during sporulation.

2 Answer to my comment 5 and the response Figure 1 indicating that the SepX function is general in *Streptomyces*, and that the *S. coelicolor* SepX is functional to complement the *S. venezuelae* Δ SepX mutant, can be included and discussed in the manuscript

Reviewer #2 (Remarks to the Author):

The authors have fully addressed all of my previous comments.

Response to Reviewer #1

1. I guess that the next sentence (line 98) needs to be rephrased (see my comment 1, and reviewer's 2 comment 2): "Microscopic analysis of this dual-labelled strain by timelapse microscopy revealed that SepX-mCherry colocalizes with FtsZ at both vegetative cross-walls and sporulation septa (Figure 1c, d, Supplementary Figure 1b and Supplementary Movie 1)".

In Figure 1b and the new Supplementary Movie 1, there are several Z-rings that do not colocalise with SepX (see examples marked with arrows in a time-lapse frame; the PDF with the image was sent to the Editor). The fact that not all Z-rings co-localise with SepX should be indicated and discussed in the manuscript.

This does not detract the results. As the authors indicate, they could not "resolve whether SepX localization precedes the arrival of FtsZ". This, together with the fact that not all Z-rings colocalise with SepX, indicates that further work will be necessary to fully understand the interaction between these two proteins. However, authors demonstrated that they colocalise, at least at some point, during sporulation.

We agree with the reviewer that further work is required to fully understand the role of SepX during cell division and this has been mentioned in line 419-422. We note that due to photobleaching of the weak SepX-mCherry fluorescence, we were only able to acquire images every 40 min. Thus, the temporal resolution of the subcellular co-localisation of SepX-mCherry and FtsZ-YPet is rather poor and it is very likely that we missed the optimal timepoint to observe a clear co-accumulation of SepX-mCherry and FtsZ-YPet. To address the reviewer's concern, we have rephrased the highlighted sentence which reads now:

"Microscopic analysis of this dual-labelled strain by time-lapse microscopy revealed that SepX-mCherry **frequently** co-localizes with FtsZ at both vegetative cross-walls and sporulation septa (Figure 1c, d, Supplementary Figure 1b and Supplementary Movie 1)."

2. Answer to my comment 5 and the response Figure 1 indicating that the SepX function is general in Streptomyces, and that the *S. coelicolor* SepX is functional to complement the *S. venezuelae* Δ SepX mutant, can be included and discussed in the manuscript.

We have added "Response Figure 1" to the manuscript as new Supplementary Figure 7b and included a brief discussion of the results (line 336-340).